# Noncanonical amino acids as doubly bio-orthogonal handles for one-pot preparation of protein multiconjugates

Yong Wang[1,4], Jingming Zhang[2,4], Boyang Han[1,4], Linzhi Tan[1], Wenkang Cai[1], Yuxuan Li[1], Yeyu Su[1], Yutong Yu[1], Xin Wang[1], Xiaojiang Duan[2], Haoyu Wang[1], Xiaomeng Shi[1], Jing Wang [1], Xing Yang [2,3] ✉ & Tao Liu [1] ✉

Genetic encoding of noncanonical amino acid (ncAA) for site-specific protein modification has been widely applied for many biological and therapeutic applications. To efficiently prepare homogeneous protein multiconjugates, we design two encodable noncanonical amino acids (ncAAs), 4-(6-(3-azidopropyl)-s-tetrazin-3-yl) phenylalanine (pTAF) and 3-(6-(3-azidopropyl)-s-tetrazin-3-yl) phenylalanine (mTAF), containing mutually orthogonal and bioorthogonal azide and tetrazine reaction handles. Recombinant proteins and antibody fragments containing the TAFs can easily be functionalized in one-pot reactions with combinations of commercially available fluorophores, radio-isotopes, PEGs, and drugs in a plug-and-play manner to afford protein dual conjugates to assess combinations of tumor diagnosis, image-guided surgery, and targeted therapy in mouse models. Furthermore, we demonstrate that simultaneously incorporating mTAF and a ketone-containing ncAA into one protein via two non-sense codons allows preparation of a site-specific protein triconjugate. Our results demonstrate that TAFs are doubly bio-orthogonal handles for efficient and scalable preparation of homogeneous protein multiconjugates.

To endow proteins with greater structural and functional diversity to speed up basic research and the development of translational applications, researchers have been striving for decades to tailor proteins by chemically modifying them with drugs, fluorophores, radio-isotopes, poly(ethylene glycol) (PEG) and other materials, nucleic acids, immunotoxins, enzymes, and so on. Such modifications have traditionally been achieved by random modifications of Cys or Lys side chains, but it is becoming increasingly clear that site-specific protein conjugates are often required for the desired applications. For example, clinical research has consistently demonstrated that the pharmacological properties of site-specific homogenous protein conjugates are superior to those of their heterogeneous counterparts[1]. Therefore,

numerous methods have been developed for preparing homogeneously modified proteins, and these methods have been extensively reviewed elsewhere[2–7].

Among them, incorporation of noncanonical amino acids (ncAAs) via genetic code expansion (GCE) is perhaps one of the best because the conjugation site can be precisely controlled with bioorthogonal chemistry that is exclusive to natural protein building blocks. The superiority of this method has recently been reinforced by the development of various site-specifically-modified protein therapeutics. For example, precise, receptor-biased IL-2 PEGylation using a copper-free click reaction of azide ncAAs incorporated at specific sites has demonstrated potential utility for immunomodulatory and antitumor

[1]State Key Laboratory of Natural and Biomimetic Drugs, Chemical Biology Center, Department of Molecular and Cellular Pharmacology, School of Pharmaceutical Sciences, Peking University, 38 Xueyuan Road, 100191 Beijing, China. [2]Department of Nuclear Medicine, Peking University First Hospital, 100034 Beijing, China. [3]Institute of Medical Technology, Peking University Health Science Center, 100191 Beijing, China. [4]These authors contributed equally: Yong Wang, Jingming Zhang, Boyang Han. ✉e-mail: yangxing2017@bjmu.edu.cn; taoliupku@pku.edu.cn

therapy[8,9]. In addition, ARX788 is a novel antibody–drug conjugate (ADC) comprising the anti-HER2 antibody trastuzumab and a potent cytotoxic drug site-specifically conjugated to the antibody via the ncAA *p*-acetyl phenylalanine (pAcF), which is genetically encoded into the antibody. Compared with non-site-specifically modified ADCs, ARX788 shows superior pharmacokinetics, safety, and antitumor activity[10,11]. Moreover, recombinant CRISPR Cas family enzymes containing genetically encoded ncAAs can be site-specifically conjugated with oligonucleotides, which are required for guide RNA conjugation based on structures, to markedly enhance the enzymes' genome editing efficiency[12–14].

However, these advances in protein modification have focused on appending only one type of functional group, and methods for grafting two or more functional groups onto proteins are urgently needed for personalized and precision medicine applications. For example, ADCs composed of only a single type of cytotoxin are at risk for the development of drug resistance, and these ADCs show low efficiency[15], which promotes the exploration of ADC with drug combinations[16]. In addition, there is a trend toward integration of both diagnostic and therapeutic functions into protein-based drugs, such as a recently reported multifunctional pretargeting strategy for cancer theranostics[17]. Thus, it is often desirable to simultaneously and site-specifically equip a single protein with combinations of molecules—including, but not limited to, radioisotopes, fluorophores, PEGs, and different drugs—to achieve research, diagnostic, and therapeutic objectives.

However, there are only a few methods for preparing homogeneous protein dual and multiconjugates, and they are often tedious and low yielding[18]. Thus, the development of a general, economical, efficient, versatile method for plug-and-play preparation of homogeneous protein multiconjugates, preferably in one-pot, for scalable translational applications would be highly desirable. As mentioned earlier, ncAA incorporation via GCE has emerged as the preferred approach for site-specific modification of proteins. Recent advances in the incorporation of two or three ncAAs have made it possible to genetically encode more than one ncAA site-specifically, thus allowing for preparation of homogeneous protein dual conjugates[19–27]. For example, the Liu group used two different bioorthogonal ncAAs bearing keto and azide groups that were incorporated into proteins in response to UAA and UAG codons with two sets of orthogonal Pyl- and TyrRS/tRNA pairs to achieve site-specific dual labeling of a protein in one pot for the first time[19]. At around the same time, Chin and colleagues used four-base codons as blank codons either alone or in combination with nonsense codons as a promising strategy for encoding two or even three distinct ncAAs into proteins[22–24,28]. In 2013, Schultz, Xiao, and colleagues used mutually orthogonal *Mb*PylRS/tRNA$_{UUA}$ and *Ec*TyrRS/tRNA$_{CUA}$ pairs to incorporate two reactive ncAAs (one with an azide side chain and one with a keto side chain) into a full-length antibody in response to UAA and UAG codons in mammalian cells, and then they functionalized the antibody with a fluorophore and a toxic drug[21]. Guo et al. achieved dual ncAA incorporation by using two different quadruplet codons[29]. The Mehl group also developed a dual encoding system to simultaneously and site-specifically install azide and tetrazine ncAAs into proteins in vivo, thus allowing simultaneous dual labeling[26]. Chatterjee et al. were the first to encode three ncAAs into a single protein: specifically, they successfully modified a protein with cyclopropene-lysine, *p*-azido-phenylalanine, and 5-hydroxytryptophan by using three mutually orthogonal aaRS/tRNA pairs that respond to three nonsense codons. The resulting protein was successfully modified by means of three mutually compatible reactions[25]. Similarly, Söll et al. developed a triply orthogonal system that uses UAG and UAA codons and a reprogrammed initiation codon UAU simultaneously to produce proteins containing three distinct ncAAs that can be separately labeled with fluorescent probes[30]. In addition, some novel ncAA-mediated

bioorthogonal reactions—such as photoinduced tetrazole–alkene cycloadditions[31,32] and reactions of an isocyano group with tetrazine substituents via [4 + 1] cycloadditions[7,33]—could potentially be used for double or multiple modifications of proteins.

However, methods for site-specifically incorporating multiple distinct ncAAs are both inefficient and operationally complex, involving multiple components and requiring tedious optimization and evaluation processes. Therefore, their scalability for therapeutic applications is limited.

In this work, we install azide and tetrazine "click" handles belonging to azide–cyclooctyne cycloaddition reaction and tetrazine–transcyclooctene (TCO) reaction pairs respectively, which are demonstrated mutually orthogonal by previous works[34,35] and density functional theory (DFT) calculations[36], into each of two designed isomeric ncAAs 4-(6-(3-azidopropyl)-s-tetrazin-3-yl) phenylalanine (pTAF) and 3-(6-(3-azidopropyl)-s-tetrazin-3-yl) phenylalanine (mTAF), and introduce the TAFs into proteins of interest by means of GCE. Subsequent conjugation by means of the two orthogonal reaction pairs enable efficient, simultaneous incorporation of two different functionalities into the encoded proteins in one pot under physiological conditions. We demonstrate that by using this system, we can combinatorially label antibody fragments (Fab and scFv) with fluorescent probes, drugs, polymers, or radionuclides. The yield of doubly modified proteins is high enough to allow in vivo experiments in various mouse models, suggesting that this platform holds promise for broadening the functional utility of proteins for both basic and translational research and that the platform can accelerate the upgrading of protein-based drugs already in clinical use.

## Results

### Design, synthesis, and genetic encoding of the bifunctional ncAA pTAF

Tetrazine–TCO reactions (inverse-electron-demand Diels–Alder [IEDDA] reactions) and azide–cyclooctyne cycloaddition reactions (strain-promoted azide–alkyne cycloaddition [SPAAC] reactions) are orthogonal[34,35]; that is, they can occur in one pot without interfering with each other. Therefore, to design our bifunctional ncAA, we chose the tetrazine and azide groups, which are small in structures. A facile and scalable synthetic route was designed to synthesize pTAF (Fig. 1a and Supplementary Fig. 1). Specifically, pTAF was synthesized by reaction of Boc-L-4-cyanophenylalanine with 4-azidobutanenitrile in the presence of hydrazine hydrate and 3-mercaptopropionic acid to form a 1,2-dihydrotetrazine, which was then oxidized and deprotected to give pTAF (Supplementary Figs. 2 and 3)[37]. This simple, and mild procedure could be carried out on a 10–20 grams scale without the need for an expensive metal catalyst[38].

Then we set out to introduce pTAF site-specifically into proteins. Guided by Tet-v2.0RS, previous developed by Mehl lab for tetrazine amino acid incorporation[39], and simulation of pTAF docking with the *Mj*TyrRS crystal structure using Schrödinger software (ver. 10.2, Schrödinger, New York, NY, USA), we designed a small focused library by fixing mutations at Q65, S158, and N162 residues around the propyl azide group, which are located within 2 Å of the propyl azide group of pTAF. Through directed evolution, we obtained a *Mj*TyrRS mutant, designated pTAFRS, that had Q65S, S158G, and N162G mutations (Fig. 1b) and efficiently incorporated pTAF into superfolder green fluorescent protein (sfGFP) containing an amber codon at Y151 (Supplementary Fig. 4). *Escherichia coli* BL21(DE3) cells carrying the reporter plasmid pET22B-sfGFP-Y151TAG and a pULTRA plasmid containing the amber suppressor pTAFRS/tRNA$^{Tyr}$ pair were induced in the absence and presence of 1 mM pTAF. A quantitative fluorescence assay revealed that fluorescence in the presence of pTAF was 80 times that in its absence (Supplementary Fig. 4). The His-tagged sfGFP containing the pTAF residue was purified by Ni-NTA affinity chromatography, and the purity was analyzed by SDS–PAGE (Fig. 1c). The yield of the purified

mutant protein was more than 90 mg/L. Correct incorporation of pTAF was confirmed by high-resolution intact-protein electrospray-ionization mass spectroscopy (Exp: 27743.96 Da, Obs: 27743.50 Da; Fig. 1d).

### Verification of mutual orthogonality and determination of rate constants for protein-labeling reactions

To verify that pTAF-containing proteins could be labeled with dibenzocyclooctyne (DBCO) and TCO simultaneously, orthogonally, and efficiently, we selected anti-HER2-scFv (referred to hereafter as HER2-scFv) from trastuzumab as a model protein. This protein comprises an N-terminal light chain variable region and a C-terminal heavy chain variable region linked by a 218-peptide linker (GSTSGSGKPGSGEGS)[40]. We screened the region near the S9, K42, and V15 residues, away from the binding site, after studying the crystal structure of HER2-scFv (Supplementary Fig. 5, PDB ID "6DN0"). The protein was produced and purified from the periplasmic space[41,42], in *E. coli* BL21 (DE3) co-transformed with pULTRA-pTAFRS and pET22b-HER2-scFv-S9/K42/V15TAG plasmids. A HER2-scFv-K42 pTAF mutant, which had a high expression yield (30 mg/L), was chosen for further characterization; and the mutant and wild-type proteins were confirmed by SDS–PAGE and high-resolution mass spectrometry (Supplementary Figs. 6 and 7). To confirm the orthogonality of tetrazine-mediated IEDDA and azide-mediated SPAAC reactions of pTAF, we chose TCO-PEG3-FITC (donor, FITC = fluorescein isothiocyanate) and DBCO-CY5 (acceptor, CY5 = Cyanine5) as reactive fluorescence resonance energy transfer probes (Fig. 2a). The following three reactions of HER2-scFv-pTAF were performed: labeling with DBCO-CY5 to generate HER2-scFv-CY5, labeling with TCO-PEG3-FITC to generate HER2-scFv-FITC, and simultaneous labeling with DBCO-CY5 and TCO-PEG3-FITC to generate HER2-scFv-CY5-FITC in one pot. The three conjugates, along with unlabeled HER2-

scFv-pTAF, were characterized by SDS–PAGE and fluorescence gel imaging (Fig. 2b). Upon excitation at 495 nm, HER2-scFv-FITC displayed a characteristic fluorescence emission at 517 nm; and upon excitation at 630 nm, HER2-scFv-CY5 displayed a characteristic fluorescence emission at 650 nm. A strong intramolecular fluorescence resonance energy transfer signal was observed only for the double conjugate, suggesting successful dual labeling. To further assess the orthogonality of the reactions, we performed high-resolution intact-protein mass spectrometry on the labeled proteins (Fig. 2c). In each spectrum, only one major peak, corresponding to the expected labeled product, was observed. From the peak intensities, the labeling efficiencies were semiquantitatively determined to be 90–95%. Within the spectra of the products obtained under the three labeling conditions described above, we did not observe any peaks for dual FITC labeling (HER2-scFv-K42-FITC-FITC, expected mass 29,058.02 Da) or dual CY5 labeling (HER2-scFv-K42-CY5-CY5, expected mass 29,390.76). Taken together, these results suggest that the one-pot tetrazine-TCO IEDDA and azide-DBCO SPAAC reactions of the pTAF-containing protein were indeed mutually orthogonal.

We next sought to determine whether the reactivity of tetrazine was affected by the adjacent conjugation with the alkyl azide. For this purpose, we used an assay based on the quenching of GFP fluorescence when tetrazine ncAAs are incorporated at Y151 and the rapid restoration of fluorescence upon reaction with conformationally strained TCO (sTCO)[43,44]. The rate constant for the reaction between sfGFP-151 tetrazine and sTCO analog sTCO-CycP-PEG2-OH were measured when the azide was unreacted and when it was occupied with DBCO-amine (Fig. 2d and Supplementary Fig. 8). The second-order rate constant for the reaction of sfGFP-Y151-pTAF with sTCO-CycP-PEG2-OH was determined to be 62,488 $M^{-1}s^{-1}$, which is consistent with the values reported for other tetrazine ncAAs[39,44]. When the azide

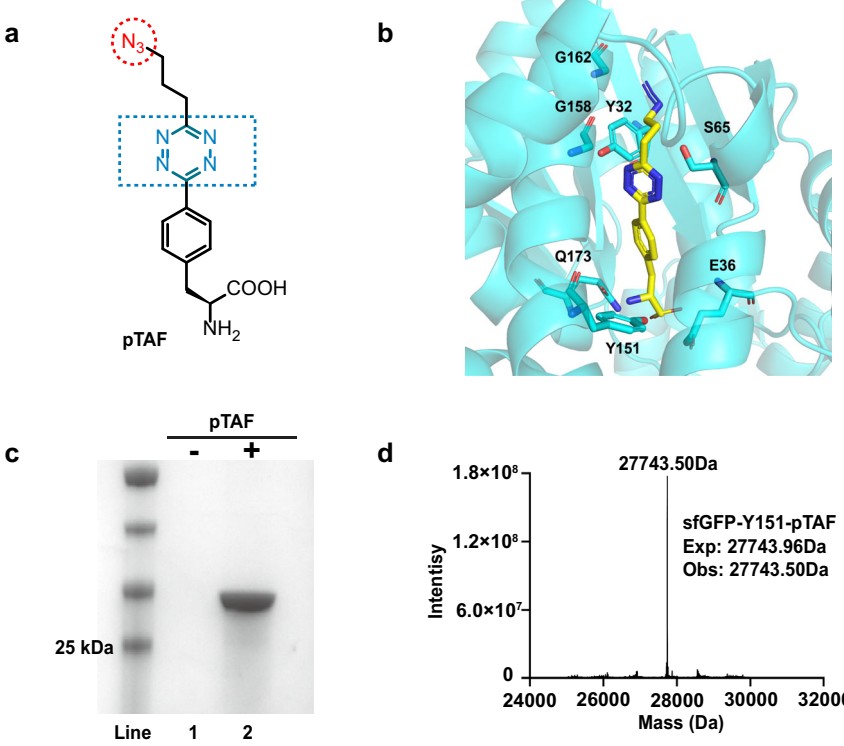

**Fig. 1 | Site-specific incorporation of pTAF into proteins. a** Chemical structure of pTAF. **b** Simulation of pTAF docking within the *Mj*TyrRS backbone (PDB ID "1J1U"; the S65, G158, and G162 residues around the propyl azide group were selected for directed evolution for pTAF incorporation. **c** SDS–PAGE analysis of site-specific incorporation of pTAF in response to UAG. Lanes 1 and 2 show the pTAF dependent

production of sfGFP-Y151-pTAF-His × 6. The purification experiments were repeated three times with reproducible results. **d** High-resolution electrospray-ionization mass spectrum of purified sfGFP-Y151-pTAF. Source data are provided as a Source Data file.

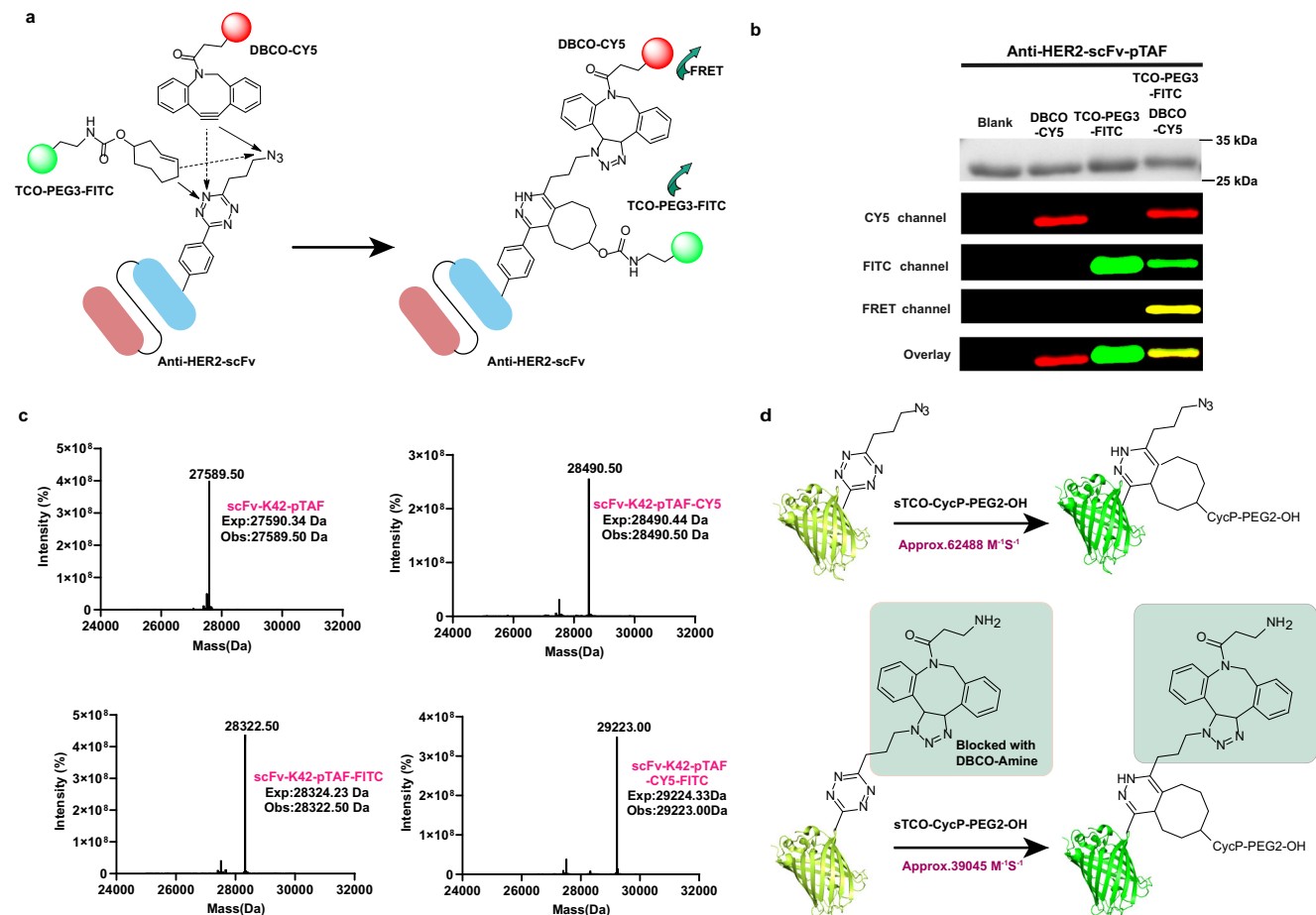

**Fig. 2 | Verification of mutual orthogonality and determination of rate constants for protein-labeling reactions. a** Schematic of dual labeling of HER2-scFv with TCO-PEG3-FITC and DBCO-CY5 via IEDDA and SPAAC reactions, respectively. **b** SDS–PAGE and the mobility shift assay of products generated by reaction of HER2-scFv-K42 pTAF with DBCO-CY5, TCO-PEG3-FITC, or both were imaged by Coomassie staining and in-gel fluorescence. TCO-PEG3-FITC excitation: 495 nm, emission: 517 nm. DBCO-CY5 excitation: 630 nm, emission 650 nm. Fluorescence resonance energy transfer excitation, 495 nm; emission: 650 nm. The purification experiments were repeated three times with reproducible results. **c** High-resolution electrospray-ionization mass spectra of scFv-K42-pTAF, scFv-K42-pTAF-CY5, scFv-K42-pTAF-FITC, and scFv-K42-pTAF-CY5-FITC. Source data are provided as a Source Data file. **d** Rate constants for reactions of sfGFP-Y151-pTAF and sfGFP-Y151-pTAF-DBCO-amine with sTCO-CycP-PEG2-OH. Source data are provided as a Source Data file.

group was preconjugated with DBCO-amine, the rate constant was 39,045 $M^{-1} s^{-1}$, which is on the same order of magnitude as the rate constant for the sfGFP-Y151-pTAF tetrazine reaction for the unreacted azide (Supplementary Figs. 9 and 10). To further rule out the possibility of steric hindrance, we have also determined the reaction rates of azide in the presence of unreacted tetrazine or tetrazine occupied after reacting with TCO-small molecule or TCO-PEG at both the free amino acid level and proteins containing pTAF residue, respectively. The results verified that no obvious steric hindrance effect on azide reactivity was observed on pTAF at both amino acid and residue levels, indicating that the propyl linker between azide and tetrazine is long enough to afford efficient one-pot dual reactions. (Supplementary Figs. 11 and 12).

### Preparation of antibody fragment near-IR dye and PEG dual conjugates for quick tumor imaging with a high signal-to-noise ratio

Near-IR fluorescence (NIRF) imaging has great potential for non-invasive imaging of tumors in vivo, and this technique has proved to be practical for achieving thorough resection of tumor tissue with fluorescence guidance during surgery[45–47]. For example, NIRF imaging targeted at prostate-specific membrane antigen (PSMA) is emerging as an attractive strategy for complete surgical removal of cancerous tissue in the prostate. Small-molecule and antibody-based bioconjugates are

popular tools for NIRF imaging, but the low molecular weight of small-molecule-based bioconjugates results in rapid blood clearance and a low signal-to-noise ratio. In addition, attachment of NIRF dyes can influence the ability of small molecules to bind to PSMA[48]. The J591 antibody is widely used to target PSMA for NIRF imaging of prostate cancer (PCa), but owing to slow clearance of the full-length antibody from the blood, high signal-to-noise ratios can be achieved only several days post-injection (p.i.), which is not convenient for clinical applications. Antibody fragments (Fab and scFv) show better tumor penetration than full-length antibodies[49], but their short blood half-lives limit their utility for NIRF imaging of tumors. Chemical modification of protein therapeutics with PEG is a commonly used strategy to increase their in vivo half-lives, and the PEG molecular weight can be fine-tuned to achieve the desired effect[50].

To enable fast NIRF imaging of prostate tumors with a high signal-to-noise ratio, we modified the J591 antibody Fab fragment with pTAF so that the fragment could be conjugated with both DBCO-CY7 and TCO-PEG20K. The dual conjugate had a theoretical molecular weight slightly higher than the cut-off size for glomerular filtration (~60 kDa)[51]. It would have the potential for enhanced tumor penetration and better in vivo pharmacokinetics. J591 Fab containing a A121 pTAF mutation, where the alanine at 121 position was mutated to ncAA pTAF by site-directed mutagenesis of alanine codon to an amber codon, was purified from the periplasmic space in *E.coli* BL21(DE3) along with the wild-

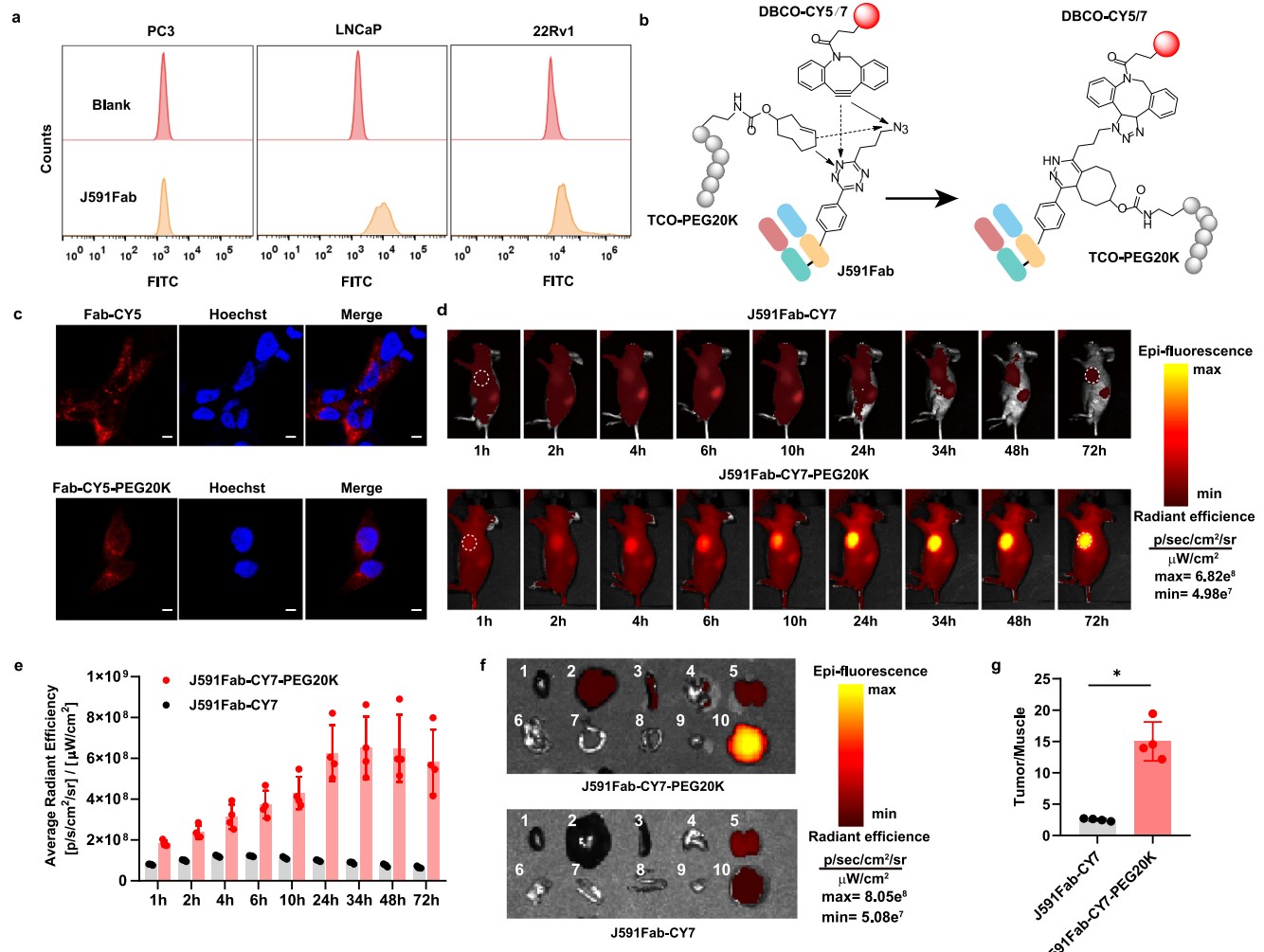

**Fig. 3 | Preparation of anti-PSMA antibody fragment near-IR dye and PEG dual conjugate for imaging of PSMA positive tumor with optimal time window in mice. a** Flow cytometry analysis of wild-type J591Fab binding to PC3 cells (PSMA−), LNCaP cells (PSMA+), and 22Rv1 cells (PSMA+). **b** Dual labeling of J591Fab with TCO-PEG20K and DBCO-CY5/CY7 via IEDDA and SPAAC reactions, respectively. **c** Confocal microscopy images of 22Rv1 cells stained with J591Fab-CY5 (red) and J591Fab-CY5-PEG20K (red). The nuclei were stained with Hoechst (blue). Scale bars = 20 μm. Assays were repeated three times with reproducible results. **d** In vivo whole-body NIRF imaging of PCa-bearing mice after intravenous injection of J591Fab-CY7 or J591Fab-CY7-PEG20K at various time points. The dotted circles indicate tumor locations. **e** Quantification of fluorescence intensity of 22Rv1 tumors in panel **d**; n = 4 mice for each group. **f** Ex vivo evaluation of dissected organs at 72 h after injection of J591Fab-CY7 and J591Fab-CY7-PEG20K. Organ labels: 1, heart; 2, liver; 3, spleen; 4, lung; 5, kidney; 6, stomach; 7, muscle; 8, small intestine; 9, large intestine; 10, tumor. **g** Tumor-to-background ratios for 22Rv1-tumor-bearing mouse in panel **f**. Muscle was used as the background tissue; n = 4 mice for each group. Mann–Whitney U test (two-sided, p = 0.029). Statistics (*p < 0.05). Data in **e** and **g** are showed as means ± SDs. Source data of **e** and **g** are provided as a Source Data file.

type protein[41]. A121 site was chosen on the basis of its surface accessibility, high expression yield and high conjugation efficiency[10]. The yields of the purified wild-type J591 Fab and the J591Fab-A121-pTAF mutant were 2 and 1.5 mg/L, respectively. The purity, reactivity, and identity of the proteins were confirmed by SDS-PAGE, fluorescence imaging, and high-resolution mass spectrometry (Supplementary Figs. 13 and 14). The activity of purified wild-type Fab was confirmed by flow cytometry analysis of PSMA-positive 22Rv1/LNCaP cells and PSMA-negative PC3 cells stained with FITC-conjugated Fabs (Fig. 3a and Supplementary Fig. 15). Next, we used CY5/CY7 dye as a labeling reagent first for the convenience of fluorescent imaging to prepare J591Fab-CY5/CY7 by reaction of the J591Fab-A121-pTAF with DBCO-CY5/CY7, and we prepared dual conjugate J591Fab-CY5/CY7-PEG20K by reaction of the J591Fab-A121-pTAF with DBCO-CY5/CY7 and TCO-PEG20K in one pot (Fig. 3b). It is worth mentioning that the protein−PEG conjugation rate was typically slow because both reactants are large; therefore, it is preferable to use TCO-modified PEG to take advantage of the fast tetrazine IEDDA reaction. The resulting

conjugates–J591Fab-CY5 were characterized by SDS−PAGE and fluorescence imaging, which indicated that the dual conjugation reaction was successful (Supplementary Fig. 16). The resulting conjugate effectively stained PSMA-positive cells (22Rv1 and LNCaP) but not PSMA-negative PC3 cells, as indicated by confocal fluorescence microscopy using J591Fab-CY5 and J591Fab-CY5-PEG20K (Fig. 3c and Supplementary Fig. 17).

To determine whether the J591Fab-CY7-PEG20K dual conjugate performed superior for in vivo tumor imaging, we selected low-to-moderate PSMA-expressing 22Rv1 tumors, which can be used to test the detection limits of the two conjugates and to model the heterogeneity of PSMA expression. Tumor-bearing mice were intravenously injected with 60 μg of J591Fab-CY7 or J591Fab-CY7-PEG20K in 100 μL of PBS and then imaged at 1, 2, 4, 6, 10, 24, 34, 48, and 72 h p.i.; four mice per group were imaged with an IVIS Spectrum Imaging System. The imaging and fluorescence intensity data showed obvious difference between the two treatment groups at all imaging time points (Fig. 3d, e and Supplementary Fig. 18). At later time points, the mice injected with

the J591Fab-CY7-PEG20K conjugate exhibited a gradually increasing fluorescence signal in the tumor regions, and the highest average radiant efficiency $(6.49 \pm 1.65) \times 10^8$ p/sec/cm$^2$/sr was reached at 48 h p.i. In contrast, the average radiant efficiency of the mice injected with J591Fab-CY7 reached a maximum $(1.22 \pm 0.02) \times 10^8$ p/sec/cm$^2$/sr at 6 h p.i., after which the protein was quickly metabolized. Ex vivo evaluation of dissected organs at 72 h p.i. revealed that fluorescence appeared mainly in the liver and kidneys, and the two groups showed a clear difference in tumor fluorescence intensity: $(0.56 \pm 0.07) \times 10^8$ p/sec/cm$^2$/sr for J591Fab-CY7 versus $(4.52 \pm 0.93) \times 10^8$ p/sec/cm$^2$/sr for J591Fab-CY7-PEG20K (Fig. 3f and Supplementary Fig. 19). As shown in Fig. 3g, the tumor-to-muscle ratio for the J591Fab-CY7-PEG20K group $(15.01 \pm 3.1)$ was approximately 6 times higher than that for the J591Fab-CY7 group $(2.52 \pm 0.20)$ at 72 h p.i. These data indicate that the attachment of PEG20K did indeed improve the immunoimaging properties of the NIR-dye-conjugated antibody fragments. Such PEG conjugates may provide a controllable imaging-time window than full-length antibodies for tumor NIRF immunoimaging in the clinic because the PEG size can be precisely fine-tuned.

## Double labeling of anti-PSMA antibody fragment for dual-modality PET and fluorescence-image-guided surgery

Dual-modality imaging can be used both for preoperative immuno-positron emission tomography (PET) imaging of the tumor lesions and for interoperatively fluorescence-image-guided identification of tumor margins during surgery. Antibodies labeled with a radionuclide and a fluorescence probe have been shown to have potential as dual-modality agents for treating PCa[52]. However, limitations imposed by the structures of the probes used in conventional and random labeling can reduce both the batch reproducibility and the bioactivity of the probes. To the best of our knowledge, no method for simultaneous site-specific labeling with a radionuclide and a fluorescence probe have been reported. To demonstrate the utility of the platform described herein, we again targeted PSMA for dual-modality imaging of PCa by conjugating J591Fab-A121-pTAF with fluorescent DBCO-CY5 and TCO-NOTA (NOTA = 1, 4, 7-Triazacyclononane-N, N′, N″-triacetic acid). The resulting conjugate was subsequently labeled with $^{64}$Cu site-specifically (Fig. 4a).

We evaluated the efficacy of sequential PET and fluorescence imaging with the dual-modality antibody fragment (Fig. 4b). Mice bearing 22Rv1 tumors were injected intravenously with 3.7–4.44 MBq $^{64}$Cu-J591Fab-CY5, and PET images were collected at various time points. PSMA positive 22Rv1 tumors were specifically visualized as early as 1 h p.i. and continued to be visible for up to 24 h p.i. (Fig. 4c and Supplementary Fig. 20). Then we also performed fluorescence imaging and imaging-guided tumor resection at 36 h p.i. The 22Rv1 tumors were clearly observed in the fluorescence images and could be completely removed. No obvious fluorescence signal was observed in the surrounding normal tissue (Fig. 4d). The resected tumor tissue was stained with anti-PSMA antibody for immunofluorescence imaging (Fig. 4e). There was good co-localization of $^{64}$Cu-J591Fab-CY5 and PSMA, which indicates the specificity of this radiotracer. In addition, we performed biodistribution studies to assess the pharmacokinetics and PSMA-specific binding of $^{64}$Cu-J591Fab-CY5; mice bearing PSMA-PC3 tumors were used as controls. Mice were injected intravenously with 148 KBq $^{64}$Cu-J591Fab-CY5, and radioactivity was detected in the organs at 12 h p.i. $^{64}$Cu-J591Fab-CY5 accumulated specifically in 22Rv1 tumors, reaching $3.50 \pm 1.87$ %ID/g, which was 2.86 times the value in PC3 tumors (Fig. 4f and Supplementary Fig. 21). The tumor-to-blood ratio was $8.14 \pm 4.15$, indicating rapid clearance from the blood pool. The high uptake of $^{64}$Cu-J591Fab-CY5 in the kidneys $(72.88 \pm 18.78$ %ID/g) indicated that the tracer was excreted mainly by the kidneys (Fig. 4f); this result is consistent with the in vivo fluorescence imaging result. The above-described results demonstrate the efficacy of

the dual-modality radiotracer $^{64}$Cu-J591Fab-CY5 and suggest that it may have the potential for further translational study.

## Site-specific antibody Fragment-PEG-Drug dual Conjugates for targeted therapy

ADCs are typically constructed by coupling tumor-targeting antibodies with potent toxin payloads via various conjugation/linker technologies. To date, fourteen approved ADCs are on the market, and hundreds of ADCs are currently in clinical trials. However, the currently available ADCs have some limitations. Conventional ADCs typically use full-length monoclonal antibodies (-150 kDa), and their size can restrict tumor penetration[53]. In addition, the bivalent binding mode can also limit deep tumor access owing to the affinity barrier effect[54]. Smaller ADCs offer the promise of more-effective delivery of drugs to solid tumors, although at the cost of more rapid clearance[55]. One of the most critical considerations is to identify the most appropriate size for maximizing the therapeutic window[56]. Antibody fragments are an obvious choice for smaller-format ADCs, and they are currently being developed as potentially better agents against solid tumors[57,58]. However, both Fab and scFv show poor pharmacokinetics because they are smaller than the glomerular filtration size cut-off and thus may require frequent injections.

To circumvent these deficiencies, we have developed a strategy for ADC design: antibody Fragment-PEG-Drug dual Conjugate (FPDC). As we demonstrated earlier, the size of antibody fragments can be fine-tuned by attaching PEGs of various molecular weights, thus providing an opportunity to precisely adjust the pharmacokinetic–pharmacodynamic relationship to achieve an optimal clinical outcome with a large therapeutic window. Using our dual-reactive amino acid pTAF, we anticipated that we could efficiently synthesize antibody fragments modified site-specifically with PEG and a drug. To evaluate our strategy, we chose HER2-positive breast cancer as a representative solid tumor model. HER2-scFv bearing pTAF at A121 was conjugated with DBCO-PEG3-VC-PAB-MMAE (DBCO-MMAE, MMAE = monomethyl auristatin E) and TCO-PEG (PEG molecular = 20 or 40 K), as described earlier (Fig. 5a, b and Supplementary Fig. 22).

To determine whether the 20 or 40 kDa PEG disturbed HER2-scFv binding to the HER2 receptor, we determined the binding affinity between Her2/ERBB2 protein and scFv-WT, scFv-pTAF-CY5, scFv-pTAF-PEG20K-CY5, and scFv-pTAF-PEG40K-CY5 by means of biolayer interferometry analysis (BLI), and the KD values were determined to be $17.0 \pm 1.6$ pM, $72.1 \pm 1.3$ pM, $0.22 \pm 0.005$ nM, and $0.45 \pm 0.007$ nM for each protein conjugate, respectively. (Supplementary Fig. 23 and Supplementary Table 1). These results indicate that although the PEGylation has an influence on the binding of scFv to HER2 receptor in a size-dependent manner, the conjugates still maintain a sub-nanomolar high binding affinity.

Next, we evaluated how PEGylation affected the internalization efficiency and distribution of the conjugates. Immunofluorescence analysis was used to observe the internalization of the conjugates through the cell membrane to the lysosome from 1 to 12 h at 37 °C (Fig. 5c and Supplementary Fig. 24). These results demonstrated that the scFv-CY5-PEG conjugates could enter the lysosome degradation pathway after internalization, suggesting that this strategy may pave the way for the development of scFv-based PEG-drug dual conjugates for tumor treatment.

To evaluate the cytotoxicity of the HER2 dual conjugates, we treated HER2-positive HCC1954 cells and HER2-negative MDA-MB-231 cells with scFv-MMAE, scFv-MMAE-PEG20K, scFv-MMAE-PEG40K, or Taxol (positive control) and carried out an MTT assay after 72 h of incubation. All the conjugates reduced HCC1954 cell viability, and the IC$_{50}$ values were calculated to be 9.7, 380.6, and 360.3 nM for scFv-MMAE, scFv-MMAE-PEG20K, and scFv-MMAE-PEG40K, respectively. Meanwhile, the conjugates showed no cytotoxicity when incubated

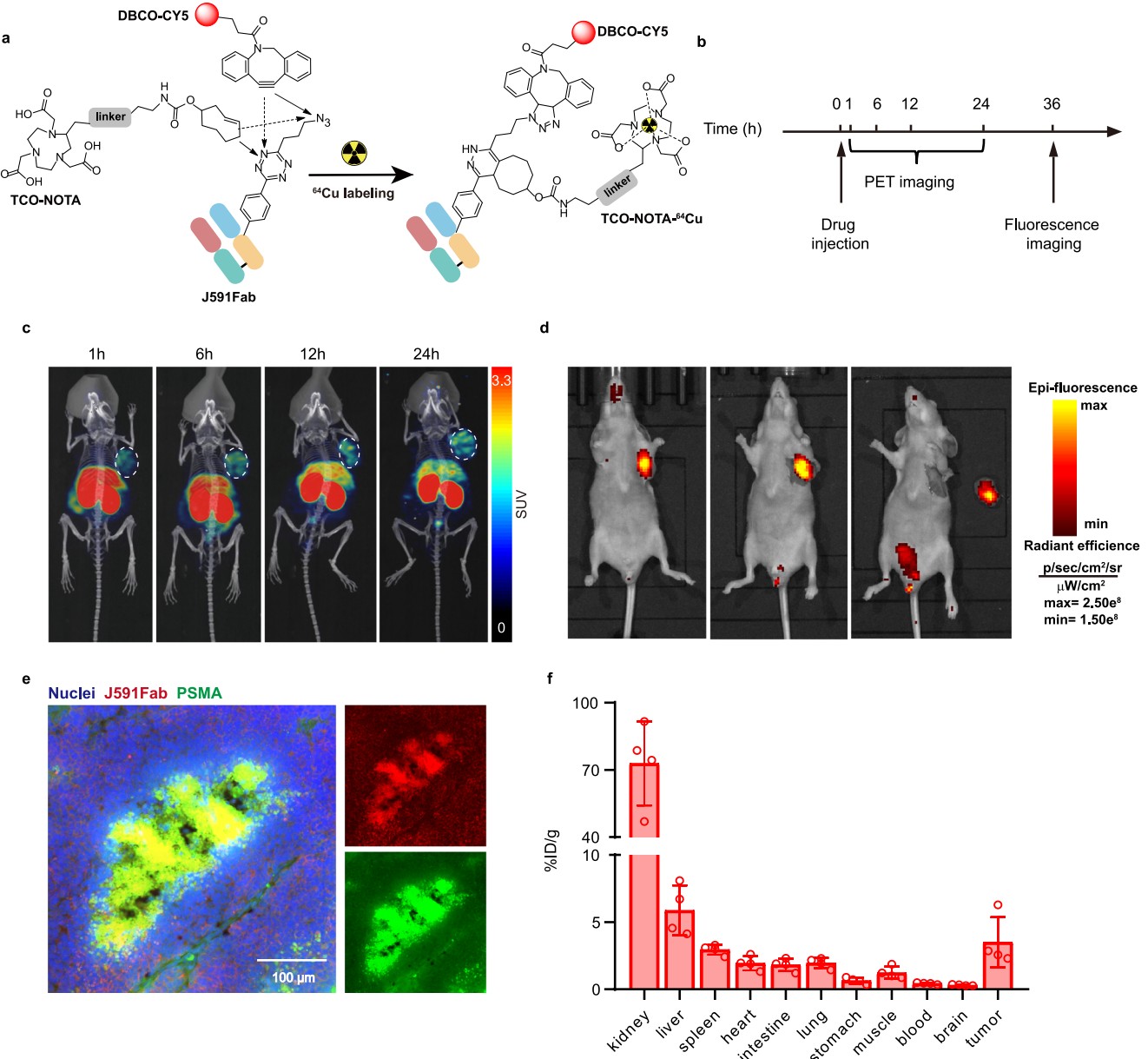

**Fig. 4 | Preparation of anti-PSMA antibody fragment dual-conjugate for micro-PET imaging and fluorescence-image-guided surgery of PSMA positive tumor in mice. a** Dual labeling of J591Fab with DBCO-CY5 and TCO-NOTA-⁶⁴Cu via SPAAC and IEDDA reactions, respectively. **b** Timing of micro-PET and fluorescence imaging of mice bearing 22Rv1 tumors. **c** Micro-PET imaging of subcutaneous 22Rv1 tumors (indicated by dashed white circles) at different time points after injection of ⁶⁴Cu-J591Fab-CY5. **d** Fluorescence-image-guided tumor dissection following PET diagnosis 36 h after injection of ⁶⁴Cu-J591Fab-CY5 at 3.7–4.44 MBq. From left to right: intact mouse, mouse after removal of the skin covering the tumor, and mouse after excision of the tumor. **e** Immunofluorescence staining of 22Rv1 tumor tissues. The tumor tissue was stained with anti-PSMA antibody (green), and the nuclei were stained with DAPI (blue); J591Fab-CY5-NOTA is shown in red. Assays were repeated three times with reproducible results. **f** Biodistribution of ⁶⁴Cu-labeled J591Fab-CY5-NOTA 12 h after injection ($n = 4$ mice for the 22Rv1 group). Data are means ± SDs. Source data are provided as a Source Data file.

with HER2-negative MDA-MB-231 cells (Fig. 5d). Although PEGylation had some effect on the cytotoxicity of the conjugates, they still showed high HER2 selectivity and potency. Taken together, our results show that antibody fragments site-specifically modified by dual-reactive pTAF could be used to prepare site-specifically PEGylated homogeneous antibody fragment-drug conjugates.

Based on the above in vitro results, we further evaluated the in vivo antitumor efficacy of antibody fragment-PEG-drug dual conjugates in HCC1954 tumor-bearing mice. The in vivo distribution of scFv-CY5 (~30 kDa) and scFv-CY5-PEG40K (~70 kDa) were first investigated via whole-body fluorescence imaging for determining the optimal dose and dosing interval. HCC1954 tumor-bearing mice were

intravenously injected with scFv-CY5 (5 mg/kg) or scFv-CY5-PEG40K (5 mg/kg) and then imaged at various time points post injection. We found that there was only weak fluorescence signal in the tumors of scFv-CY5 group. However, tumors in scFv-CY5-PEG40K group could be clearly visualized at the later imaging time points (Fig. 6a), indicating the accumulation of PEG-drug dual conjugates in the tumor site. Quantitative data showed that the fluorescence intensity of tumors in the scFv-CY5 group reached its peak at only 2 h and then decreased rapidly. By contrast, in the scFv-CY5-PEG40K group, the fluorescence signal of tumors increased throughout the imaging time points (Fig. 6b). The maximum intensity of tumors in scFv-CY5-PEG40K group (~$2.10 \times 10^9$ p/sec/cm²/sr at 72 h) was 5.13 folds higher than that in scFv-

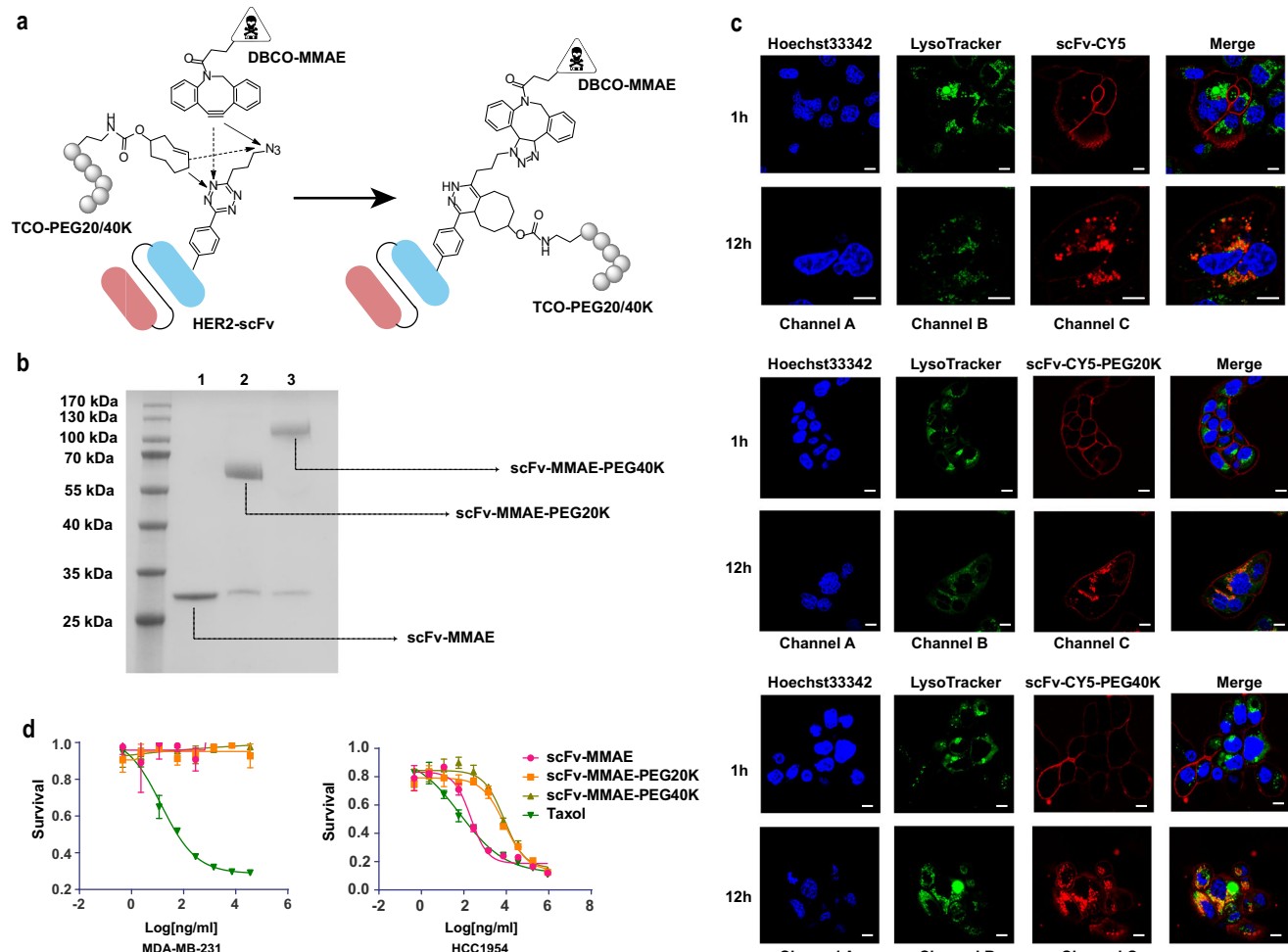

**Fig. 5 | Preparation of antibody fragment-PEG-drug dual conjugate. a** Synthesis of an antibody fragment-PEG-drug dual conjugate prepared with DBCO-MMAE and TCO-PEG20K or TCO-PEG40K via SPAAC and IEDDA reactions, respectively. **b** SDS−PAGE analysis of scFv-MMAE, scFv-MMAE-PEG20K, and scFv-MMAE-PEG40K (lanes 1–3, respectively). The purification experiments were repeated three times with reproducible results. **c** Assessment of scFv-CY5, scFv-CY5-PEG20K, and scFv-CY5-PEG40K internalization. HCC1954 breast cancer cells were treated with the three CY5 bioconjugates (red), incubated for 1 or 12 h, and then stained with LysoTracker (green) and Hoechst 33342 (blue). Visualization of lysosome stained with LysoTracker (Channel B). Visualization of the three CY5 bioconjugates label (Channel C). Overlay of Channel B and C, indicating co-localization of the three CY5 bioconjugates and LysoTracker (Merge). Scale bar = 10 μm. Assays were repeated three times with reproducible results. **d** In vitro MTT assay of the cytotoxicity of the conjugates to HCC1954 and MDA-MB-231 cells The $IC_{50}$ values of Taxol alone were 39 and 10 nM for HCC1954 and MDA-MB-231, respectively; the values for scFv-MMAE, scFv-MMAE-20K, and scFv-MMAE-40K against HCC1954 were 9.7, 380.6, and 360.3 nM, respectively. Data are shown presented as mean ± s.d ($n = 3$ independent experiments). Source data are provided as a Source Data file.

CY5 group ($\sim$0.41 × 10$^9$ p/sec/cm²/sr at 2 h). Ex vivo imaging of tumors and major organs harvested 72 h p.i. further confirmed the enhanced tumor retention of scFv-CY5-PEG40K compared to scFv-CY5 (Fig. 6c), which was also validated by the quantitative result using the region-of-interest analysis (Supplementary Fig. 25).

Encouraged by the fluorescence imaging result in vivo, we further evaluated the antitumor efficacy of anti-HER2 FPDC containing MMAE and PEG40K in HER2-positive tumor model. Mice bearing HCC1954 tumors were injected intravenously with scFv-MMAE and scFv-MMAE-PEG40K at the dose of 10 mg/kg, and the PBS treatment group was set as negative control. Under our experimental conditions, the scFv-MMAE-PEG40K treatment group exhibited marked inhibition on tumor growth, while no tumor growth inhibition was observed in scFv-MMAE treatment group and the PBS treatment group (Fig. 6d). To further confirm the results, tumors were excised and weighted at the end of the experiments, the result also proved the superior therapeutic effect of scFv-MMAE conjugated with PEG40K (Fig. 6e). Moreover, no significant changes in body weights and liver function were observed after these treatments (Fig. 6f and Supplementary Fig. 26), indicating the good biocompatibility of our antibody fragment-PEG-drug dual

conjugates system for promising anti-tumor activity. Although in our proof-of-concept, the tumor was not completely eliminated, we think it can be improved by adjusting the dose and injection frequency.

## Preparation of site-specific protein triconjugates with dual ncAAs incorporation and three mutually compatible bioorthogonal reactions

Inspired by the high incorporation efficiency described above, we wondered whether pTAF could be combined with other bioorthogonally reactive ncAAs to enable efficient triple labeling of proteins. Because ketone-mediated oxime ligation is fully orthogonal to the tetrazine-mediated IEDDA reaction and the azide-mediated SPAAC reaction[20,59], we chose pAcF, which was used to prepare an ADC that is currently in a clinical trial[60], as a second ncAA to pair with pTAF. *Mj*TyrRS and PylRS pairs are commonly used for mutually orthogonal incorporation of two ncAAs in response to two nonsense codons, and we selected one such pair for dual incorporations. Unfortunately, the conditions for incorporation of pAcF and pTAF were not orthogonal. Therefore, we designed and synthesized mTAF (Supplementary Figs. 27, 28, and 29 and Fig. 7a), which could potentially be recognized

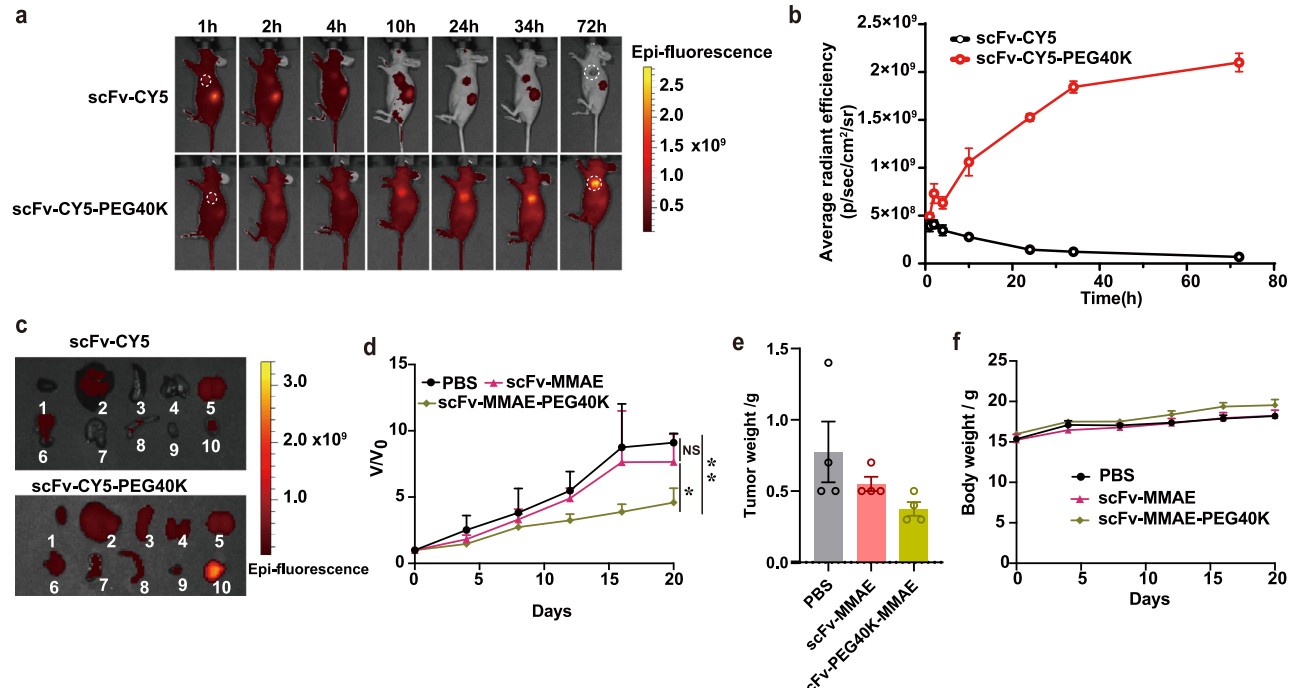

**Fig. 6 | In vivo antitumor activity of antibody fragment-PEG-drug dual conjugates. a** In vivo fluorescence imaging of HCC1954 tumor-bearing mice after intravenous injection of scFv-CY5 (top) or scFv-CY5-PEG40K (bottom) at various time points. The dotted circles indicate tumor locations. **b** Quantification of fluorescence intensity of HCC1954 tumors in panel a; $n = 3$ mice for each group. **c** Ex vivo evaluation of dissected organs at 72 h after injection of scFv-CY5 (top) and scFv-CY5-PEG40K (bottom). Organ labels: 1, heart; 2, liver; 3, spleen; 4, lung; 5, kidney; 6, stomach; 7, small intestine; 8, large intestine; 9, muscle; 10, tumor. **d** Relative tumor volume (the ratio of tumor volume to initial size before treatment) curves of tumor as a function of time treated with PBS, scFv-MMAE (10 mg/kg) and scFv-MMAE-PEG40K (10 mg/kg) with equivalent amounts of MMAE. **e** Weight of the excised tumors after treatments above. **f** Body weight change of mice after intravenous administration of PBS or the conjugates. Kruskal–Wallis test ($p = 0.018$) and Tukey HSD test (PBS vs scFv-MMAE, $p = 0.356$; PBS vs scFv-MMAE-PEG40K, $p = 0.004$; scFv-MMAE vs scFv-MMAE-PEG40K, $p = 0.033$). Statistics (\*$p < 0.05$; \*\*$p < 0.01$). Data are presented in panels **b** ($n = 3$), **d**, and **f** as mean ± SD ($n = 4$) and in panel **e** as mean ± SEM ($n = 4$). Source data of **b**, **d**–**f** are provided as a Source Data file.

by evolved *Mb*PylRS for genetic incorporation[44,61]. mTAF was synthesized in 70–80% overall yield from Boc-L-3-cyanophenylalanine. By screening a small collection of synthetase mutants, we determined that R2-74RS/tRNA$_{CUA}$, which was evolved by the Mehl group for Tet-v3.0 incorporation[44], efficiently recognized mTAF. We then verified that mTAF could be incorporated into sfGFP at the 151 TAG position (Fig. 7b and Supplementary Fig. 30), which was accomplished with an expression yield of up to 75 mg/L. Incorporation was also confirmed by intact-protein mass spectrometry (Fig. 7c). Because PylRS pairs can be used in mammalian expression systems, we also assessed whether mTAF could be incorporated into HEK293T cells by co-transfecting the cells with a pCMV-R2-74RS-mTAFS/tRNA$_{CUA}$ plasmid and a pcDNA3.1(-)-mCherry-linker18-LeuTAG-eGFP reporter plasmid (Supplementary Fig. 31). This experiment indicated that mTAF could indeed be used to label proteins that require a eukaryotic expression system, such as full-length antibodies, cytokines and even viruses.

To simultaneously incorporate two ncAAs (Fig. 7d), we constructed two synthetase pairs, R2-74RS/tRNA$_{UUA}$ and pAcFRS/tRNA$_{CUA}$, into pULTRA[62] and pEVOL plasmids, respectively, and used them to co-transform *E. coli* BL21 (DE3) cells containing a pET22b-sfGFP-51TAA-151TAG vector[63]. We showed that full-length sfGFP could be efficiently produced only in the presence of both mTAF and pAcF (Supplementary Fig. 32). Next, we constructed HER2-scFv-S9TAG-K42TAA and HER2-scFv-V15TAG-K42TAA plasmids and produced double mutant proteins in a manner similar to that described above. The yields of both doubly mutated proteins were about 20 mg/L, which is about 70% of the wild-type protein yield and is high enough for the follow-up experiments. The two ncAAs substitutions were then confirmed by high-resolution intact-protein mass spectroscopy (Fig. 7e and Supplementary Figs. 33 and 34).

With the HER2-scFv-pAcF-mTAF mutant in hand, we attempted to label the protein with hydrazide-CY5, DBCO-CY3, or TCO-PEG20K separately, with pairs of the labels, or with all three labels in a catalyst-free one-pot reaction. All reactions were performed at room temperature for 10 h except for reactions involving hydrazide-CY5, which were slow for 3 days. The resulting conjugates above were characterized by fluorescence gel imaging analysis. The analysis showed that corresponding conjugates was accompanied by the matching fluorophore and molecular weight (20 K PEG) shift with minimal cross-reaction (Fig. 7f). These results demonstrated that a dual incorporation and triple labeling strategy could be used to achieve efficient site-specific triple labeling of a protein by incorporation of both pAcF and mTAF.

## Discussion

Chemical modification of proteins not only is important for basic research but also is starting to play a critical role in the development of protein-based therapeutics. Site-specific protein conjugates have repeatedly demonstrated overall pharmacological profiles that are superior to those of their heterogeneous counterparts. However, as mentioned above, the few available methods for preparing homogeneous protein dual conjugates are often tedious and low yielding. Of the existing methods for site-specific protein modification, ncAA incorporation via GCE is state-of-the-art because the stoichiometry and the site of conjugation can be precisely controlled with bioorthogonal chemistry. Applications of this method have recently been encouraged by clinical trials (NCT04829604 and NCT04009681) of protein therapeutics containing site-specific chemical modifications achieved with genetically encoded ncAAs. However, as the need increases for multifunctionalized proteins—e.g., proteins bearing combinations of

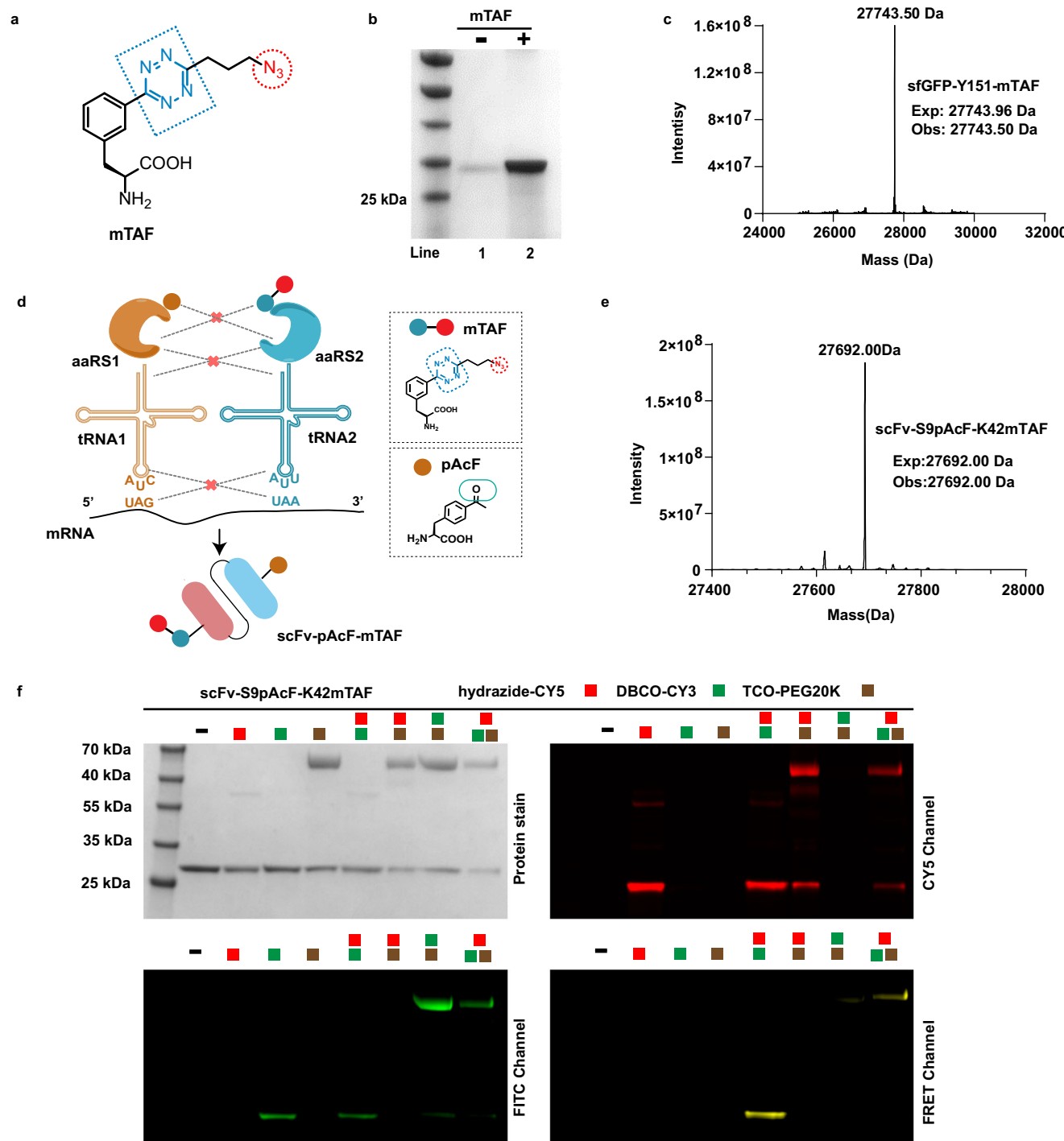

**Fig. 7 | Site-specific dual incorporation of mTAF and pAcF into proteins for triple modification. a** Structure of mTAF. **b** SDS-PAGE analysis of site-specific incorporation of mTAF in response to UAG. Lanes 1 and 2 show mTAF-dependent production of sfGFP-Y151-mTAF-His×6. The purification experiments were repeated three times with reproducible results. **c** High-resolution mass spectrum of a purified sfGFP mutant containing mTAF mutant (Expected Mass 27743.96 Da, Observed Mass 27743.50 Da). **d** Schematic of strategy for encoding two ncAAs with two sets of orthogonal aaRS/tRNA pairs, which suppress the UAG and UAA stop codons to drive incorporation of pAcF and mTAF into proteins. **e** High-resolution mass spectrum of purified mutant scFv containing both mTAF and pAcF mutants (Expected Mass 27692.00 Da, Observed Mass 27692.00 Da). **f** SDS-PAGE of scFv-S9pAcF-K42mTAF after reaction with combinations of hydazide-CY5, DBCO-CY3, and TCO-PEG20K; images were obtained by Coomassie staining and in-gel fluorescence through chemoselective reactions. Assays were repeated three times with reproducible results. Source data of **c** and **e** are provided as a Source Data file.

drugs, fluorophores, radioisotopes, PEG/materials, nucleic acids, immunotoxins, enzymes—the need for methods that allow efficient, controllable, and scalable production of proteins with two or more modifications will become urgent. Although site-specific incorporation of two distinct ncAAs is possible for protein dual labeling, the yields are

lower than those for incorporation of a single ncAA and are too low for in vivo therapeutic applications.

Here we demonstrate that ncAAs bearing mutually orthogonal tetrazine and azide groups on the side chain can be synthesized. The resulting dual-reactive ncAAs (TAFs) can be genetically encoded and

efficiently and site-specifically incorporated into recombinant proteins in both *E. coli* and mammalian cells. We show that under the labeling conditions we test, the labeling reactions are orthogonal to each other, and that the reaction of the azide group with DBCO only minimally perturbs the tetrazine reactivity toward TCO. Both chemistries are compatible with commercially available protein-modification reagents. The conjugation reactions can be completed within a few hours in one pot, and no products generated by double labeling of a single substrate are found, confirming that the reactions are indeed mutually orthogonal. Using antibody fragments as model proteins, we demonstrate that we can simultaneously and site-specifically incorporate combinations of commercially available labels, including radioisotopes (for early tumor diagnosis through PET imaging), fluorophores (for surgical navigation of tumors via fluorescence imaging), PEG (for improved pharmacological properties), and drugs (for targeted cancer therapy). We demonstrate that the yields of purified doubly conjugated proteins are high enough to allow for in vivo experiments in mouse models to explore the above-mentioned applications.

Using this platform, we site-specifically label antibody fragments, such as anti-HER2-scFv and anti-PSMA-Fab, with various combinations of fluorescent probes, drugs, PEG, and radionuclides. First, we accomplish high-resolution tumor imaging with Fab-PEG-NIRF conjugates. Second, we demonstrate that anti-PSMA-Fab conjugated with PET/fluorescence labels allows dual-modality imaging for early diagnosis of PCa and subsequent image-guided surgery. Third, we conjugate drugs and PEG chains of various lengths to anti-HER2-scFv to generate small format FPDC targeting HER2-positive breast cancer cells, as a potentially promising strategy to achieve deep tumor penetration. Finally, we develop a system for simultaneously incorporating two ncAAs, i.e., mTAF and pAcF, into a single protein in response to two nonsense codons. This system affords a site-specifically triple-labeled protein conjugate in relatively good yield.

In summary, we develop a convenient, versatile platform that allows easy access to protein dual and multiconjugates in high yields by site-specific incorporation of ncAAs containing two mutually orthogonal and bioorthogonal reactive groups by means of GCE. This platform can easily be used to expand the functional utilities of proteins with commercially available protein-modification reagents, thus making it an economical option for studying the consequences of attaching various combinations of functional molecules to proteins of interest. This platform can be expected to be particularly valuable for expanding the scope of protein-based therapeutics and generating therapies, such as combinations of various drugs conjugated to antibodies that can overcome the limitations of currently available immunotherapies.

## Methods

### Ethical statement
All procedures and protocols were approved by the Animal Ethics Committee at Peking University Frist Hospital (Beijing, China), approval number: J202261. The maximum tumor volume allowed by the Ethics Committee of the Peking University Frist Hospital was 2000 mm$^3$ per tumor, which was not exceeded in these studies.

### General information
All common laboratory reagents were purchased from commercial sources. All reactions were monitored by HPTLC Silica Gel 60 GF254, 2.5 × 7.5 cm glass support, 0.25 mm coating, and column chromatography was carried out with Column Chromatography Silica Gel, 200−300 mesh purchased from Leyan, China. NMR data were acquired on a Bruker 400 spectrometer and chemical shifts (δ) for $^1$H and $^{13}$C were reported in p.p.m. scale using solvent residual peak as an interior label. High-resolution ESI mass data were obtained on the Thermo Q

Exactive HF-X instrument. The protein molecular weight was analyzed by Waters Synapt G2-Si (Waters, Manchester, UK).

### 4-azidobutanenitrile (2)
4-bromobutyronitrile (**1**, 10.0 g, 68.0 mmol) was dissolved in DMSO (150.0 mL), and sodium azide (5.7 g, 87.8 mmol) was added to the solution. The reaction mixture was stirred at 80 °C for 12 h. After the reaction had completed, the mixture was poured into water, and extracted with diethyl ether. The combined organic layer was washed with brine, dried over Na$_2$SO$_4$, filtered and concentrated under reduced pressure to afford 4-azidobutanenitrile (**2**) as a light yellow oil[64] (6.0 g, yield 80.2%).

### (S)−3-(4-(6-(3-azidopropyl)−1,2,4,5-tetrazin-3-yl) phenyl)−2-((tert-butoxycarbonyl) amino) propanoic acid (4, Boc-pTAF)
Boc-L-4-cyanophenylalanine (**3**, 1.0 g, 3.5 mmol), 4-azidobutanenitrile (**2**, 3.0 g, 27.3 mmol) and 3-mercaptopropionic acid (0.4 g, 3.4 mmol) were added to a flask and cooled to 0 °C. To this mixture was added dropwise hydrazine hydrate (2.7 g, 54.4 mmol). The reaction mixture was stirred at 40 °C for 12 h under argon protection. After the reaction had completed, the mixture was poured into water and cooled to 0 °C. Sodium nitrite (3.6 g, 52.5 mmol) was dissolved in water and added to the mixture. Then 1 M hydrochloric acid was added dropwise until the pH reached 2.0-3.0. The mixture was extracted with ethyl acetate and the combined organic layer was washed with brine, dried over Na$_2$SO$_4$, filtered and concentrated under reduced pressure. The residue was purified by silica column (DCM: MeOH = 30:1) to afford Boc-pTAF (**4**) as a purple solid (1.0 g, yield 68.0%). ESI-MS: (m/z): [M-H]$^-$: 427.1844.

### (S)−2-amino-3-(4-(6-(3-azidopropyl)−1,2,4,5-tetrazin-3-yl) phenyl) propanoic acid (pTAF)
Boc-pTAF (**4**, 1.0 g, 2.3 mmol) was dissolved in 20% TFA in DCM and stirred at room temperature until the starting materials were consumed monitoring by TLC. After completion, remove the solvent under reduced pressure, added 10 mL diethyl ether and filtered resulting in a purple color solid of pTAF (0.7 g, yield 91.3%). $^1$H NMR (400 MHz, CD$_3$OD) δ 10.30 (1 H, s), 8.53 (2 H, m), 7.62-7.64 (2 H, m), 3.53 (2 H, m), 3.45 (3 H, m), 2.70 (2 H, m), 2.26 (2 H, m). $^{13}$C NMR (175 MHz, CD$_3$OD) δ 173.89, 169.54, 164.02, 133.26, 132.83, 129.69, 128.48, 126.86.9, 50.40, 39.03, 33.34, 32.91, 31.56, 26.69. ESI-MS: (m/z): [M + H]$^+$: 329.133.

### (S)−3-(3-(6-(3-azidopropyl)−1,2,4,5-tetrazin-3-yl) phenyl)−2-((tert butoxycarbonyl) amino) propanoic acid (6, Boc-mTAF)
Except for the starting reagent Boc-L-3-cyanophenylalanine (**5**, 2.0 g, 6.9 mmol), the other steps for the synthesis of Boc-mTAF (**6**, 2.0 g, yield 68.0%) were the same as described for the synthesis of Boc-pTAF.

### (S)−2-amino-3-(3-(6-(3-azidopropyl)−1,2,4,5-tetrazin-3-yl) phenyl) propanoic acid (mTAF)
Except for the starting reagent Boc-mTAF (**6**, 2.0 g, 4.7 mmol), the other steps for the synthesis of mTAF (1.4 g, yield 91.3%) were the same as described for the synthesis of pTAF. $^1$H NMR (400 MHz, MeOD) δ 8.53 (s, 1 H), 8.51−8.44 (m, 1 H), 7.68−7.58 (m, J = 4.5 Hz, 2 H), 3.99−3.89 (m, 1 H), 3.56 − 3.50 (m, 2 H), 3.49−3.41 (m, 3 H), 3.25−3.17 (m, 1 H), 2.33−2.21 (m, 2 H). $^{13}$C NMR (400 MHz, d6-DMSO) δ 169.45, 164.08, 139.65, 134.10, 132.28, 129.88, 129.10, 126.17, 115.83, 55.97, 50.39, 37.48, 31.84, 26.90. ESI-MS: (m/z): [M - H]$^-$: 327.1338.

### General plasmid construction
All plasmids were constructed by Gibson assembly using Hieff Clone Plus Multi One Step Cloning Kit (YEASEN, Shanghai, China). The wild-type sfGFP, J591Fab, HER2-scFv, and the mutated sfGFP (Y151TAG), sfGFP (G51TAA and Y151TAG), J591Fab (A121TAG/K169TAG/S202TAG), HER2-scFv (S9TAG/K42TAA), HER2-scFv (S9TAG and K42TAA) and IFN-

α2b (E42TAG) were cloned into the pET22b vector (Invitrogen). Proteins genes containing His × 6 tag gene were under the control of isopropyl β-D-1-thiogalactopyranoside (IPTG) inducible T7 promoter. pTAFRS (Y32G, Q65S, F108S, Q109D, S158G, and N162G) and R2-74RS were cloned into pULTRA vectors (Plasmid #48215, addgene). Meanwhile, pAcFRS[10], was also cloned into pEVOL vector (Plasmid #31186, addgene). All mutants above were generated by PCR and verified by sequencing in genewiz. These primers used in this article were listed in Supplementary Table 2.

**Cloning pTAFRS and R2-74RS into pULTRA.** RS genes for pTAF and R2-74RS were amplified with pULTRA-pTARS-FW, pULTRA-pTAFRS-RV, pULTRA-R2-74RS-FW and pULTRA-R2-74RS-RV primers. pULTRA plasmid vector was amplified with pULTRA-vec-FW and pULTRA-vec-RV. The PCR products were gel purified and the desired synthetase genes fragments were ligated into pULTRA vector by Gibson assembly. The ligated product was sequenced to confirm identity. These primers used here were listed in Supplementary Table 2.

**Cloning R2-74RS into pCMV.** R2-74RS genes were amplified with pCMV-R2-74RS-FW and pCMV-R2-74RS-RV primers. pCMV plasmid backbone from pCMV-MbPylRS(DiZPK) (Plasmid #91706, Addgene) were digested with the restriction enzymes HindIII and BamHI. The PCR products and digestion products were gel purified and the desired synthetase genes fragments were ligated into pCMV vector by Gibson assembly. The ligated product was sequenced to confirm identity. These primers used here were listed in Supplementary Table 2.

**Cloning mCherry-linker18-Leu-eGFP into pcDNA3.1 vector.** mCherry-linker18-Leu-eGFP genes were amplified with eGFP-flag-HindIII-RV and mCherry-NheI-FW primers. pcDNA3.1 plasmid backbone from pcDNA3.1(+) eGFP (Plasmid #129020, Addgene) were digested with the restriction enzymes HindIII and NheI. The PCR products and digestion products were gel purified and the desired genes fragments were ligated into pcDNA3.1 vector by Gibson assembly. The ligated product was sequenced to confirm identity. These primers used here were listed in Supplementary Table 2.

**Cloning pAcFRS into pEVOL.** pAcFRS and corresponding tRNA genes were amplified with NdeI-pAcFRS-FW and XhoI-tRNA-RV primers. pEVOL plasmid backbones from pEVOL-pAzF (Plasmid # 31186, Addgene) were digested with the restriction enzymes NdeI and XhoI. The PCR products and digestion products were gel purified and the desired genes fragments were ligated into pEVOL vector by Gibson assembly. The ligated product was sequenced to confirm identity. Then pEVOL vector bearing p15A, araC and araBAD were amplified with NheI-FW and araBAD-OP-RV, and pAcFRS gene was amplified with OP-pAcFRS-FW and saII-pAcFRS-RV. The PCR products were gel purified and underwent overlap PCR to get a fragment of P15A-arac-araBAD-pAcFRS. The ligated pEVOL plasmids above were digested with the restriction enzymes NheI and saII. The overlap PCR products and digestion products were gel purified and the fragments were ligated into pEVOL plasmids by Gibson assembly. The ligated product was sequenced to confirm identity. These primers used here were listed in Supplementary Table 2.

**QuikChange Site-Directed Mutagenesis of plasmids.** pET22b-sfGFP plasmid bearing Y151TAG mutation was obtained by QuikChange PCR. pET22b-sfGFP-WT was amplified by sfGFP-151TAG-FW and sfGFP-151TAG-RV primers. Other site-directed mutagenesis of plasmids was referred to this method.

**Proteins expression and SDS−PAGE analysis**
*E. coli* BL21 (DE3) cells, co-transformed with pET22b-protein (ampicillin resistance) and pULTRA-aaRS (spectinomycin resistance) or/and

pEVOL-aaRS (chloramphenicol resistance), were grown in 2YT medium at 37 °C. When OD600 reached 0.6, IPTG (0.5 mM) and respective ncAAs (1 mM) were added to induce the expression of proteins. *E. coli* cells, which were grown for an additional 16 h at induction temperature, and harvested by centrifugation at 2000 × *g* for 30 min. *E. coli* pellets were resuspended in lysis buffer (50 mM Tris-HCl, 200 mM NaCl, 10 mM imidazole, pH 8, 0.1 mg/ml lysozyme) and incubated at 4 °C for 1.5 h on a shaker. Affinity purification of the cytoplasmic or periplasmic and SDS−PAGE analysis was performed referring to our previous work by standard procedures[41]. LabPAGE 12% gel was purchased from Beijing LABLEAD BIOTECH Co., Ltd. (Beijing, China).

**Antibody fragments conjugation**
All conjugations were performed in PBS (pH 7.4) after the Fab or scFv fragments with TAFs were concentrated to 50 μM using a 10-kDa cutoff centrifugal filter device (Millipore). Depending on the kind of modifications we need, different reagents were added. For modification of hydrazide-CY5, antibody fragments, hydrazide-CY5 (Cat No. BDC-42, purchased from Confluore, 10 mM in DMSO, 50 eq) and aniline (0.1 eq) were added to conjugation buffer (6.5 mM $KH_2PO_4$, 95 mM $K_2HPO_4$, 125 mM NaCl, pH=8.4) and incubated at 37 °C for 3 days. For other modifications, DBCO-CY5 (Cat No. BCD-34, purchased from Confluore, 10 mM in DMSO, 4 eq), DBCO-CY7 (Cat No. BCD-22, purchased from Confluore, 10 mM in DMSO, 4 eq), DBCO-CY3 (Cat No. BCD-30, purchased from Confluore, 10 mM in DMSO, 4 eq), DBCO-(PEG)3-VC-PAB-MMAE (Cat No. HY-111012, purchased from MedChemExpress, 30 mM in DMSO, 4 eq), TCO-PEG20K (Cat No. BGNH-45, purchased from Confluore, 20 mM in ddH2O, 4 eq) and TCO-PEG40K (Cat No. BGNH-46, purchased from Confluore, 10 mM in ddH2O, 4 eq), TCO-PEG3-NOTA (Cat No. R-056, purchased from Ruixibio, 100 mM in DMSO, 4 eq) was added to the solution of protein in one pot. The solution was incubated for 4 h at 25 °C. Upon completion, unreacted compound was removed by using a 10-kDa cutoff centrifugal filter device (Millipore) or Superdex 75 Increase 10/300 GL column (Cytiva). The conjugates in PBS were stored at 4 °C for short-term use and −80 °C in aliquots for long-term use. The concentration was determined by measuring the absorbance at 280 nm and Pierce™ BCA Protein Assay Kit (Thermo Fisher Scientific).

**Measuring reaction rates of sfGFP-Tet-v3.0 with sTCO-CycP-PEG2-OH**
Fluorescence of 200−800 nM of purified sfGFP-pTAF diluted in 200 μL PBS was measured (488 nm excitation, 528 nm emission). Then 1 μL of DBCO-amine (20 mM in DMSO) was added and incubated at 25 °C for 1 h, and the fluorescence was measured to obtain the standard curve of sfGFP-pTAF and sfGFP-pTAF-amine. To determine the kinetic constants between tetrazine and sTCO-CycP-PEG2-OH (Cat No. BCT-32, purchased from Confluore, 10 mM in DMSO), 400−1600 nM of sfGFP-pTAF or sfGFP-pTAF-amine was mixed with equimolar sTCO-CycP-PEG2-OH respectively, and fluorescence was measured until no fluorescence increase was observed. Curves were fit using GraphPad Prism (v8.0.2) software to determine kinetic constants.

**Cell lines and mouse models**
LNCaP (Cat No. SCSP-5021), 22Rv1 (Cat No. SCSP-5022), and PC3 (Cat No. SCSP-532) human prostate cancer cell lines were purchased from the Chinese Academy of Sciences Typical Culture Collection (Shanghai, China). The human breast cancer cell lines HCC1954 (Meisen, China, Cat No. CTCC-003-0205) and MDA-MB-231 (Meisen, China, Cat No. CTCC-001-0019) were obtained as a generous donation from Dr. Yu Cao (Peking University Shenzhen Graduate School). HEK293T cells (Cat No. CRL-11268) were purchased from ATCC. LNCaP cells were maintained in RPMI 1640 medium containing 10% fetal bovine serum (FBS), 1% penicillin−streptomycin, 1% GlutaMax-I, and 1% sodium pyruvate. 22Rv1, PC3 and MDA-MB-231cells were maintained in RPMI 1640

medium containing 10% fetal bovine serum (FBS) and 1% penicillin–streptomycin. HCC1954 cells were maintained in RPMI 1640 medium containing 20% fetal bovine serum (FBS), 1% L-glutamine, and 1% penicillin–streptomycin. HEK293T cells were cultured in Dulbecco's Modified Eagle Medium (DMEM) supplemented with 10% (v/v) fetal bovine serum. All the cells were cultured at 37 °C under 5% $CO_2$ in the air. All animal experiments were conducted following regulations on laboratory animals of the Beijing municipality. BALB/c nude mice (Female or male, 5–6 weeks old) were obtained from the Animal Center at the Peking University Frist Hospital. Mice were group-housed (up to five mice in one cage), maintained at 20–25 °C, 40–60% room humidity and a 12 h light/dark cycle. Before further experiments, all mice were acclimatized for at least 7 days.

### Binding selectivity detected by flow cytometry
PC3, LNCaP, and 22Rv1 cells were collected using Trypsin. The cells were incubated with 200 μL blocking buffer (PBS, 1% (v/v) FBS, pH 7.4) on ice for 1 h, then washed with 200 μL ice-cold PBS twice. NHS-FITC labeled Fab was diluted to 10 μg/ml and incubated with the cells on ice for 1 h. After washing with 200 μL ice-cold PBS three times, the cells were analyzed using a CytoFlex Flow Cytometer (Beckman Coulter, CA, USA) and the data were analyzed with FlowJo software (Version 10.8.1, FlowJo, Ashland, OR, USA).

### Confocal fluorescence imaging of PCa cell lines with Fab-CY5 and Fab-PEG20K-CY5
LNCaP, 22Rv1, and PC3 cells were seeded in the 8-well chamber at a density of $5 \times 10^4$ per well and incubated for 48 h for adhesion. Cells were fixed with 4% paraformaldehyde, permeabilized with 0.5% Triton-X100 and blocked with 3% BSA. Then the cells were incubated overnight with 5 μg/mL Fab-CY5 or Fab-PEG20K-CY5. Finally, Hoechst 33342 was used for counterstaining. The images were scanned with a FluoView1000 confocal microscope (Olympus, Japan).

### In vivo fluorescence imaging of mice bearing 22Rv1 prostate tumors
PSMA positive 22Rv1 tumors were induced on 4–5 weeks old male BALB/c nude mice by subcutaneous injection of $5 \times 10^6$ cells in suspension in 100 μL PBS buffer, respectively. The mice were randomized into 2 groups ($n = 4$) for in vivo imaging when their xenografts reached approximately 500 mm³. Mice were intravenously administrated with 60 μg Fab-CY7 or Fab-CY7-PEG20K in 100 μL PBS buffer, and then the images were obtained at 1 h, 2 h, 4 h, 6 h, 10 h, 24 h, 34 h, 48 h, and 72 h by the IVIS Spectrum Imaging System (Caliper Life Sciences, Hopkinton, MA). The tumor area was delineated with a bright field and its fluorescence intensity was quantified using Living Image 4.3.1 software. The ipsilateral muscle was also delineated and set as background.

### 64Cu labeling
[64]Cu was produced at Peking University Cancer Hospital using the HM-20 medical cyclotron (20 MeV, Sumitomo, Japan). [64]Cu labeling was performed by incubating 80 μL of [64Cu] $CuCl_2$ containing 92.5 MBq activity, 65 μL of 0.1 M sodium acetate, and 600 μg (12 nmol) of Fab-CY5-NOTA at 37 °C for 30 min at pH 4.0–5.0. The PD-10 column was used to purify the radiotracer. The radiochemical purity was determined by radio-TLC to be over 95% ($n = 3$).

### Micro-PET imaging of 64Cu labeled Fab-CY5-NOTA
22Rv1 tumor-bearing mice ($n = 4$) were intravenously injected with 3.70–4.44 MBq [64Cu] Fab-CY5-NOTA in 100 μL PBS buffer. Before the corresponding imaging time, mice were anesthetized with 2.5 L/min of isoflurane and placed into a PET scanner (Super-Nova®, PINGSHENG, Shanghai, China). PET/CT imaging was performed at 1 h, 6 h, 12 h, and 24 h post-injection. Images were analyzed using the Avatar software

(version 1.6.6.5) after reconstruction of CT-AC PET with 3D-ordered subset expectation maximization (OSEM) + point spread function (PSF) algorithm.

### Bio-distribution of 64Cu-labeled Fab-CY5-NOTA
22Rv1 and PC3 tumor-bearing mice were intravenously injected with 148 KBq [64Cu] Fab-CY5-NOTA in 100 μL PBS buffer. The mice were sacrificed at 12 h post-injection. Organs of interest were dissected and weighed. The radioactivity was measured using a γ-counter (Packard, Meriden, CT) and represented as the percentage uptake of injected dose per gram of tissue (%ID/g).

### PSMA immunofluorescence of resected specimens
After IVIS imaging, 22Rv1 tumors were dissected and embedded in an optimal cutting temperature compound (OCT). Then the tumor tissues were cut into 20 μm thick slides and fixed with acetone at 4 °C for 20 min. After being blocked by BSA at room temperature for 1 h, the slides were incubated with a primary antibody against PSMA (ab19071; clone YPSMA-1, Abcam, Cambridge, UK; 1:300 dilution) at 4 °C overnight, and then stained with a FITC-labeled secondary antibody (ZF-0312; ZSGB-Bio, Beijing, China;1:150 dilution) at room temperature for 1 h. Finally, DAPI was used for counterstaining. Vectra Polaris was used to scan the stained slides.

### Biolayer interferometry analysis
BLI experiments were performed on an Octet RED96e instrument (ForteBio Inc., Sartorius, Germany) using amine-reactive AR2G biosensors. Sensor tips were hydrated for 10 min prior to use. The sensors were then activated with a freshly prepared mixture of 90 mM EDC and 25 mM sulfo-NHS for 5 min, loaded with 20 μg/mL HER2/ERBB2 Protein (10004-H02H, Sino Biological, China) in 10 mM sodium acetate (pH 4.5) for 5 min, and then excess reactive esters were quenched with 1 M ethanolamine, pH 8.5 for 5 min. Amine-coupled HER2/ERBB2 Protein was then used to capture a panel of scFvs (scFv-WT, scFv-CY5, scFv-PEG20K-CY5 and scFv-PEG40K-CY5) in kinetics buffer (PBS with 0.05% Tween20). Association was monitored for 5 min and dissociation was monitored for 20 min. Each kinetics experiment included the analyte scFv in kinetics buffer at several different concentrations, as well as a reference biosensor, loaded with HER2/ERBB2 Protein, which was kept in kinetics buffer without antibody as a zero concentration. Octet BLI Analysis 12.2 software was used for the analysis of the binding curves.

### Cell viability assay
Cells (MDA-MB-231 and HCC1954) were seeded in a culture-treated 96-well clear plate (3,000 cells per well in 100 μL culture medium) and incubated at 37 °C under 5% $CO_2$ for 24 h. Serial dilution of Taxol, scFv-MMAE, scFv-MMAE-PEG20K, and scFv-MMAE-PEG40K were added to the cells at concentrations ranging from 0.02 nM to 30 μM in each well and the plate was incubated at 37 °C for 72 h, and cell viability was determined by MTT assay. 20 μL of MTT (5 mg/ml in phosphate-buffered saline) was added to each well followed by incubation at 37 °C for 4 h. The formazan crystal sediments were dissolved in 150 μL DMSO and the absorbance at 570 nm was measured using a plate reader. $IC_{50}$ values were calculated using Graph Pad Prism 8 software. All assays were performed in triplicate.

### Real-time imaging of HER2-mediated internalization
HCC1954 cells were seeded on glass bottom dishes (29 mm dish with 20 mm bottom well, $9 \times 10^4$ cells per dish) and were incubated at 37 °C under 5% $CO_2$ for 24 h. Cells were incubated with scFv-CY5, scFv-CY5-PEG20K, and scFv-CY5-PEG40K at 20 μg/ml at 37 °C for gradient time respectively. Cells were washed with PBS to remove unbound antibodies. Hoechst 33342 (10 μg/ml, Beyotime Biotechnology) was added to the culture medium to stain cell nuclei for 30 min. Cells were washed with PBS again to remove free Hoechst dye. Then Lyso-Tracker Green

(50 nM, Beyotime Biotechnology) was added to stain lysosome for 30 min. Finally, cells were washed with PBS again and a fresh culture medium was added. Coverslips were examined using a Zeiss LSM880 Confocal microscope.

### In vivo fluorescence imaging of BALB/c nude mice bearing HCC1954 tumors

HER2-positive HCC1954 tumors were induced on 4–5 weeks old female BALB/c nude mice by subcutaneous injection of $5 \times 10^6$ cells in suspension in 100 μL PBS buffer, respectively. The mice were randomized into 2 groups ($n = 3$) for in vivo imaging when the volume of their xenografts reached ~300 mm³. Mice were intravenously administrated with 100 μg scFv-CY5 or scFv-CY5-PEG40K in 100 μL PBS buffer, and then the images were obtained at 1, 2, 4, 10, 24, 34, and 72 h by the IVIS Spectrum Imaging System (Caliper Life Sciences, Hopkinton, MA). The tumor area was delineated with a bright field and its fluorescence intensity was quantified. After the final fluorescence image acquired, the mice were sacrificed and organs of interest were collected and imaged ex vivo.

### In vivo antitumor activity of antibody fragment-PEG-drug dual conjugates

HER2-positive HCC1954 tumors were induced on female BALB/c nude mice by subcutaneous injection of $5 \times 10^6$ cells in suspension in 100 μL PBS buffer. When the tumor volume reached ~100 mm³, they were randomized into 3 groups ($n = 4$) for the following experiments. The mice were intravenously injected with PBS, scFv-MMAE (10 mg/kg) or scFv-MMAE-PEG40K (10 mg/kg) every 4 days for a total of 5 times. The tumor volumes and body weights were recorded every four days. The tumor volume was calculated according the formula: length × width² × 0.5. On day 20 post first injection, blood samples (200 μL) were collected from the mice of three groups and the serum was separated by centrifugation at $700 \times g$ for 15 min. The levels of alanine aminotransferase (AST) and aspartate aminotransferase (ALT) in serum were quantified by AST and ALT activity assay kits (Nanjing Jiancheng Bioengineering Institute), respectively. The mice were then euthanized, the tumors were excised and weighed.

**Cell culture and transfection.** pCMV-R2-74RS-mTAFS/tRNA$_{UUA}$ and pcDNA3.1(-)-mCherry-linker18-LeuTAG-eGFP plasmids were transfected by using polyethylenimine (PEI) to HEK293T cells. In brief, $1 \times 10^5$ cells were seeded in each well of 24-well plates 8 h before transfection. Cells were incubated with a transfection solution containing 3 μg PEI (Polysciences, Catalog No. 23966) and 1 μg plasmids. After 4 hours, the media was replaced with fresh media with or without 100 μM mTAF. Then the culture was incubated for 48 hours.

**Statistics and reproducibility.** All data are presented as means ± SEM or means ± SD as depicted in the figure caption. Mann–Whitney $U$ test was used to compare the differences between two independent samples. Kruskal–Wallis test and Tukey HSD test were used to compare the differences between three or more independent samples. All above statistical analysis was performed with SPSS 26.0. Statistical significance was considered at $P < 0.05$.

Statistical tests for significance were described in individual figure legends. All results were repeated by at least three biological replicates unless specified in figure legends. No statistical method was used to predetermine sample size. No data were excluded from the analyses. The experiments were randomized for the animal grouping. The investigators were not blinded to allocation during experiments and outcome assessment.

### Reporting summary

Further information on research design is available in the Nature Portfolio Reporting Summary linked to this article.

## Data availability

The high-resolution electrospray-ionization mass data, determination of rate constants, quantification of fluorescence intensity, biodistribution of $^{64}$Cu-labeled antibody fragment in vitro and in vivo, and cytotoxic measurements generated in this study, as well as uncropped or processed gels are provided in the Source Data file. Protein structures with accession numbers "1J1U" and "6DN0" were used in this work and obtained from the RCSB Protein Data Bank. The remaining data generated or analyzed in this study are available within the article and Supplementary Information. Source data are provided with this paper.

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

## Acknowledgements

This work was financially supported by National Natural Science Foundation of China (NO. 92156025, T.L.; NO. 22207002, Y.W.; NO. U22A20332, T.L.; NO. 92253301, T.L.; NO. 22277002, X.Y.; NO. 92059101, X.Y.; and NO. 21877004, X.Y.), the National Key Research and Development Program of China (NO. 2022YFA0912403, T.L. and NO. 2021YFA0909900, T.L.), Beijing Natural Science Foundation (NO. JQ20034, T.L.), and Clinical Medicine Plus X–Young Scholars Project (NO. PKU2020LCXQ029, T.L.), Peking University. The China Postdoctoral Science Foundation, No.70 General Fund, 2021 (NO. 2021M700294, Y.W.).

## Author contributions

T.L., Y.W., and X.Y. conceived and designed the research. Y.W., B.H., and W.C. performed the chemical synthesis. Y.W., B.H., X.W., Y.Y., Y.S., and L.T. performed plasmid construction, protein purification, cell culture, MTT assay, confocal microscopy imaging, Flow cytometry analysis and analytical assays. J.Z. performed mouse experiments, including mouse imaging, tumor dissection and antitumor activity in vivo. Y.L., B.H., and Y.W. performed determination of rate constants experiments. B.H., and J.W. performed biolayer interferometry experiments. X.S. performed the High-resolution electrospray-ionization mass analysis. X.D. and J.Z. performed the labeling of J591Fab-CY5-NOTA with $^{64}$Cu.; H.W. performed the experiment of pTAF docking within the *Mj*TyrRS backbone. Y.W., J.Z., and B.H. analyzed the data. Y.W., J.Z., and T.L. wrote and reviewed the paper. All authors read and approved the paper.

## Competing interests

The authors declare no competing interests.
