## [Peer Review File · Nature Communications]

REVIEWER COMMENTS

Reviewer #1 (Remarks to the Author):

Previous methods for preparing site-specific homogenous multi-conjugates have suffered from issues such as low yield and lack of generality. In this study, Wang et al. designed, synthesized, and genetically incorporated two bifunctional ncAAs pTAF & mTAF for facile protein duo modifications. The TAFs contained mutually orthogonal azide and tetrazine bioorthogonal handles, allowing one pot biofunctionalization with great reaction kinetics. The authors have demonstrated several biomedical regimens in vivo applications utilizing TAFs such as NIR fluorescence tumor imaging, cancer theranostics & imaging-guided surgery, and targeted therapy. TAFs enable general, efficient, scalable preparation of protein multi-conjugates, which is especially valuable for expanding the scope of designing protein-based therapeutics. The promising results in this study are convincing and well supported by experimental evidence and should be of interest to the general readership of Nature Communications. I recommend the acceptance of the manuscript after considering the following suggestions:

1. The confocal imaging data in Figure 5c show the internalization of the scFvs. However, the co-localization of the scFv & the lysosome decreased as the length of the conjugated PEG increases, with scFv-CY5 showing the best co-localization and scFv-CY5-PEG40K the worst. Is it possible that the PEG might delay the entry of lysosome degradation pathway? Does the co-localization get better if the incubation time increases?
2. TAFs are great for various in vitro & in vivo applications and could be especially promising for in vivo duo labeling purposes. We wonder how stable the ncAAs are under physiological conditions. A time course stability test under physiological conditions would be nice.
3. Authors should further polish the manuscript. For example, the abstract should be shortened and focused on the current limits of incorporation of multiple bioorthogonal ncAAs.
4. Figure 2, “naked N3” should be replaced with Azide structure. It is more clear
5. The authors chose the tetrazine and azide to design the bifunctional ncAA groups since tetrazine-TCO reactions and azide-cyclooctyne cycloaddition reactions are orthogonal. A brief explanation about why they are orthogonal should be given.

6. Authors chose A121 of J591 Fab for the incorporation of pTAF. The authors should explain why this residue was chosen.

7. The authors wrote that ^{64}Cu is labeled after the conjugate J591Fab-A121-pTAF with DBCO-CY5 and TCO-NOTA is formed, which does not match Fig. 4a showing that the TCO- ^{64}Cu is used in the first place instead of formed later. Also, the superscript of copper in Fig. 4a is in the wrong position.

8. The molecular weight of the marker should be labeled in Figure 5b, at least for the ones have close MW to the protein band.

9. Figure 2c & 6b the “scfv” should be changed to “scFv” to be consistent with the others in the paper

10. In line 524, there an extra “)” should be removed.

11. In line 668, a space between “CY5” and “(10 mM” is needed.

12. The figure quality is not good. High resolution figure is required.

13. “pTAF and mTAF” in the abstract should be spelled out.

Reviewer #2 (Remarks to the Author):

The paper is devoted to the production and application of protein multiconjugates using a novel type of noncanonical amino acid (ncAA), which simultaneously contains in its chemical structure two bioorthogonal groups for site-specific labelling (azide and tetrazine). It has been shown that such ncAAs can be incorporated into the protein sequence using a genetic code expansion (GCE) technology and expressed proteins,

The authors present a new strategy for bioorthogonal protein labeling via an incorporation into a protein a newly synthesized ncAA, which is equipped with 2 functional groups taking part in

bioorthogonal reactions: azide group – in strain-promoted alkyne-azide cycloaddition (SPAAC) and tetrazine group – in inverse electron-demand Diels-Alder reaction (iEDDA). GCE technology was used to introduce such modified ncAA into the protein. Developed ncAA could provide an advantage in GCE efficiency, since it requires the incorporation of only one ncAA instead of two nCAAs, bearing each only one bioorthogonal handle; and afterwards higher GCE efficiency leads to the increase in protein yield. Newly produced bi- and tri- conjugates were further labelled with dyes, radioisotopes and drugs in a “one-pot” manner and interesting applications were shown.

The introduction summarizes in detail and at length different strategies for making dual ncAA conjugates and highlights the disadvantages of dual encoding, which for GVVE typically requires two orthogonal RS pairs and suffers from lower expression yield. These are valid points that their method does not necessarily suffer from.

However, the introduction does not mention that dual functional group encoding nCAAs have been encoded more than once before. E.g. see *ChemBiochem*. 2017 Jan 17;18(2):181-184 and *Biochemistry*. 2018 Aug 7;57(31):4747-4752. Here, two other functional groups were encoded, one for labelling and for crosslinking, but from the type of journal published, it is also clear that not any combination of two known functionalities (of over 200 encoded ones) into one ncAA will depict a substantial advance. So overcoming the actual drawback of having two functional groups in close proximity (i.e. on one side chain) should be an important point to address for the readership of a broad interdisciplinary journal like *Nature communication*. The fact that no large linker can be encoded between two functional groups with current RS systems, and thus steric hindrance effects can become a concern, is a disadvantage that needs to be clearly shown to outcompete the disadvantages from dual ncAA incorporation with the discussed methods.

In that context, much more convincing needs to be done to make this relevant for the readership of *Nature com* in my eyes, such as

- unfortunately, no side-by-side experiment has been demonstrated to compare how the presented approach outperforms a strategy with incorporation of 2 nCAAs. However it would be important to see such results, keeping in mind that new binding pockets for 2 synthetases (MjTyrRS and MbPylRS) was evolved for new nCAAs and efficiency of tRNA aminoacylation can be diminished.

- probable steric hindrance that might appear when bulky compounds are selected for protein modification. Steric hindrance theoretically can cause a decrease in a reaction rate or even absence of reaction when labelling group is shielded by a compound, which has been first covalently attached to a protein. Authors provide data in Fig. 2d, where they show that reactivity of tetrazine group is not compromised after reaction between DBCO-amine and azide group, however iEDDA reaction is several orders of magnitudes faster than SPAAC and tetrazine group will react first. Thus, it would be interesting to see the opposite experiment where tetrazine group has already reacted with a corresponding compound via iEDDA and then assess a reaction rate of SPAAC between azide group and complementary compound. If reaction rate of SPAAC is decreasing substantially, it might be a problem for conjugate production.

I have two more major concerns, that are easier addressed but currently lower my enthusiasm for the paper.

- Cy5 is a red fluorescent dye, and not an IR dye. I checked some marketing flyers from some producing companies, and even those do not claim Cy5 to be near IR. Cy5.5 is the first dye that is considered by those companies with self-interest as near IR, and I would argue that from a visible spectrum point of view, the first proper IR dye from the cyanine derivatives is Cy7.

- I appreciate the breath of experiments done for important applications of multi conjugates for diagnostic and therapy and thus make the paper very interesting to read indeed, However, the statistics in many and in particular the mice experiments are low and I understand that one needs to be considerate with number of mice sacrificed and I do not suggest to do more of those in the interest of living mice. However, at the same time, how this is sold needs to be also careful. The students t-test is not even defined for a sample size of 3 or 4, and the use of proper statistical tools or more careful wording is certainly needed when advertising the work.

Suggested further improvements

- The statement in line 447: "Taken together, our results show that antibody fragments site-specifically modified by dual-reactive pTAF could be used to prepare homogeneous fragment-PEG-drug dual conjugates" need its own set of analytics to validate the homogenous claim.

-

- Statement line: 268 filtration (~60 kD)⁴⁹. It was small enough for enhanced tumor penetration and it had a faster serum clearance rate than full-length antibodies" needs reference or experiment.

- Page 5. "anhydrous hydrazine, which is ... and not commercially available". I would recommend to omit this detail here.

- Page 6. "218-peptide linker (GSTSGSGKPGSGEGS)". Please cite a paper, from which sequence of linker was taken. At least in Whitlow M. et al, Protein Eng., 1993 the sequence of 218-peptide linker is GSTSGSGKPGSGEGSTKG..

- Figure S5. In the caption it is mentioned that protein migrates as a single band at about 27 kDa, but 2 bands are clearly visible on the gel. Please use accurate description and do you have any assumptions why 2 bands appeared?

- Fig. 1c and S5 uses different style: "Kd" and "kDa". Please unify the style of labeling.

- Fig. S8 – please insert a title of the caption

- Fig. S11. Why was FRET channel omitted in this figure? For other figures demonstrating results for in-gel fluorescence this channel was demonstrated.

- In Fig. 3d another bright focus of dye accumulation rather than the tumor is visible. Does it belong to accumulation in kidneys or stomach? Interestingly, the brightest organ after dissection seems to be a stomach according to Fig. 3f.. Fig. S15 shows that side dye accumulation happens preferentially in kidney and stomach, while text says: "Ex vivo evaluation of dissected organs at 72 h p.i. revealed that fluorescence appeared mainly in the liver and kidneys". Please fix the text to avoid misunderstanding.

- “J591 Fab containing a pTAF mutation at A121..” That is not clear what “pTAF mutation” is. Please rephrase this sentence.

- “The apparent binding affinities (EC50) of scFv-CY5, scFv-CY5-PEG20K, and scFv-CY5-PEG40K were determined to be 1.163, 7.316, and 23.93 μ M, respectively (Fig. S19). These results indicate that the conjugates had high binding affinity for the HER2 receptor, although the PEGylation did have influence the binding in a size-dependent manner.” Measured binding affinities do not seem to be high, taking also into account that Her2-scFv with higher nanomolar Kd have been published. Did you try another method to check and calculate binding affinity of your wild-type Her2-scFv?

- Fig. 5C. It seems not to be any colocalization between lysotracker and scFv-CY5-PEG40K, while text says: “These results demonstrated that the scFv-CY5-PEG conjugates could enter the lysosome degradation pathway after internalization”. Please fix the test or describe more in details.

- “All the conjugates reduced HCC1954 cell viability, and the EC50 values were calculated to be 9.7, 380.6, and 360.3 nM for scFv-MMAE, scFv-MMAE-PEG20K, and scFv-MMAE-PEG40K, respectively”. Probably authors meant here IC50, but not EC50. Please fix.

- “The maximum intensity of tumors in scFv-CY5-PEG40K group ($\sim 2.10 \times 10^9$ p/sec/cm²/sr at 72h) was 5.13 folds higher than that in scFv-CY5 group ($\sim 4.09 \times 10^8$ p/sec/cm²/sr at 2h).” It would be better to use the same extent for values to make more clear 5.13-fold difference.

- Fig. S21 – ALT and AST abbreviations should be explained, they might not be obvious for all readers.

- Fig. 7b. Gel shows a dim, but clear band for “-ncAA”. Do you suppose to have leakage in orthogonality there? Did new synthetase start recognizing endogenous AA? Did you do any negative selection to prevent recognition of endogenous ncAA?

- Fig. 7d. Sequence in blue tRNA should be UUA as mentioned in a text. Please fix a mistake.

- Fig. S26. Quite bright band can be observed for condition “- pAcF + mTAF”, so it seems that MjTyrRS starts recognizing mTAF for tRNA aminoacylation. At the same time there are also dim bands already in conditions “- pAcF - mTAF” and “+ pAcF – mTAF”. This evidence of leakage in orthogonality should be mentioned somewhere in a text or in the caption.

- Fig. S27. Please precise what you mean with “-ncAA”: absence of 1 or 2 ncAAs.

The authors would like to thank the reviewers for precise and thoughtful comments and constructive criticism which has led to a better manuscript. Herein, we address the comments as follows.

Comments from Review #1:

Previous methods for preparing site-specific homogenous multi-conjugates have suffered from issues such as low yield and lack of generality. In this study, Wang et al. designed, synthesized, and genetically incorporated two bifunctional ncAAs pTAF & mTAF for facile protein duo modifications. The TAFs contained mutually orthogonal azide and tetrazine bioorthogonal handles, allowing one pot biofunctionalization with great reaction kinetics. The authors have demonstrated several biomedical regimens in vivo applications utilizing TAFs such as NIR fluorescence tumor imaging, cancer theranostics & imaging-guided surgery, and targeted therapy. TAFs enable general, efficient, scalable preparation of protein multi-conjugates, which is especially valuable for expanding the scope of designing protein-based therapeutics. The promising results in this study are convincing and well supported by experimental evidence and should be of interest to the general readership of Nature Communications.

Reply:

We really appreciate your recognition of our work!

I recommend the acceptance of the manuscript after considering **the following suggestions:**

1. The confocal imaging data in Figure 5c show the internalization of the scFvs. However, the co-localization of the scFv & the lysosome decreased as the length of the conjugated PEG increases, with scFv-CY5 showing the best co-localization and scFv-CY5-PEG40K the worst. Is it possible that the PEG might delay the entry of lysosome degradation pathway? Does the co-localization get better if the incubation time increases?

Reply:

Thank you for the careful review of our paper. To address your question, we performed a time course experiment to determine the effect of PEGylation on internalization. The result is shown in Figure R1. Exactly as you anticipated, PEGylation indeed delay the entry of lysosome degradation pathway and yes, the co-localization gets better when the incubation time increases. The best co-

localization signal for scFv-CY5-PEG20/40K was observed when we extended the incubation time to 8-12 hours. These data have been included in the revised Figure 5c and Figure S22.

Figure R1. The confocal merge imaging showing the internalization of the scFv conjugates at different time points. Co-localization of scFv-CY5, scFv-CY5-PEG20K, and scFv-CY5-PEG40K with lysosome were imaged by merging pictures of CY5 staining and Lyso-Tracker Green staining. HCC1954 cells were seeded on glass bottom dishes (29 mm dish with 20 mm bottom well, 9×10^4 cells per dish) and were incubated at 37 °C under 5% CO₂ for 24 h. Then the cells were incubated with different scFv-CY5 conjugates at 20 µg/ml at 37°C for gradient time. Culture medium containing Hoechst 33342 (10 µg/ml) and Lyso-Tracker Green (75 nM) was added to the cells to stain cell nuclei and lysosome respectively for 30 min. Cells were washed with PBS to remove free dye and fresh culture medium was added. Glass bottom dishes were examined using Zeiss LSM880 Confocal microscope. Scale bars = 10 µm.

2. TAFs are great for various in vitro & in vivo applications and could be especially promising for in vivo duo labeling purposes. We wonder how stable the ncAAs are under physiological conditions. A time course stability test under physiological conditions would be nice.

Reply:

To address your question, we first assessed the stability of pTAF & mTAF in phosphate buffer saline (PBS, pH 7.4) and undiluted fetal bovine serum (FBS), respectively. The ncAAs were incubated in PBS or FBS for 0-72 hrs, respectively, followed by LC-MS analysis using *O*-methyltyrosine as an internal standard. The time-percentage (% amount of initial TAFs is defined as 100%) profiles for pTAF & mTAF were calculated and shown in Figure R2. Data shows that around 90% of both pTAF & mTAF remain intact in PBS for 72 h at 37 °C (Figure R2a, 2b) and in FBS for 24 h at 37 °C (Figure R2c, 2d). The amount of intact ncAAs for both TAFs reached around 60% when incubation time increased to 72 h at 37 °C in FBS (Figure R2c, 2d). These data suggest that both ncAAs are stable in PBS solution, and are moderately stable under physiological conditions for potential *in vivo* applications, considering that the half time of most small molecules are less than a few hrs.

Figure R2. **Time-percentage (%) of pTAF & mTAF stability in PBS (a, b) and undiluted FBS (c, d) at 37 °C for 0-72 h.** Three independent trials were averaged to time-percentage (%) using the standard deviation as the error. Note: No error bars appear for some points because of they are shorter than the size of the symbol.

Next, we evaluated the stability of TAFs that were encoded into proteins. IFN- α 2b containing pTAF at E42 on hand was selected as our model protein. The IFN- α 2b-E42-pTAF was incubated in the PBS (pH 7.4) for 36 hours at 25 °C and 37 °C. The stability of TAF residue was assayed by conjugating with DBCO-CY5 and/or TCO-PEG3-FITC and analyzed using SDS-PAGE mobility shift assay. As shown in Figure R3, IFN- α 2b-E42-pTAF maintained full bio-orthogonal reactivity for both reactions, suggesting a high stability in PBS buffer for protein dual conjugation.

Figure R3. **SDS-PAGE mobility shift assay showing the stability and reactivity of pTAF encoded into protein.** IFN- α 2b-E42-pTAF was incubated in PBS buffer (pH 7.4) for 36 hours at 25 °C and 37 °C.

3. Authors should further polish the manuscript. For example, the abstract should be shortened and focused on the current limits of incorporation of multiple bioorthogonal ncAAs.

Reply:

Thanks for the suggestion. We have re-described the abstract, highlighted the research focus, and polished the language and grammatical errors throughout the manuscript.

4. Figure 2, “naked N₃” should be replaced with Azide structure. It is more clear

Reply:

“Naked N₃” have been replaced with Azide structure in Figure 2d according to your suggestion.

5. The authors chose the tetrazine and azide to design the bifunctional ncAA groups since tetrazine–TCO reactions and azide–cyclooctyne cycloaddition reactions are orthogonal. A brief explanation about why they are orthogonal should be given.

Reply:

Thank you for the suggestion. The orthogonality has been reported in the literature. Hilderbrand et al. demonstrated that the bulky cyclooctynes (DBCOs) react efficiently with azides (k_2 : 1-60 M⁻¹s⁻¹) but not with tetrazines and TCO react efficiently with tetrazine (k_2 : 10²-10⁶ M⁻¹s⁻¹) but not with azides (Angew. Chem. Int. Ed. 2012, 51, 920–922). The mechanistic interpretation of the orthogonality has also been reported (J. Am. Chem. Soc. 134, 17904–17907). Meanwhile the orthogonality of tetrazine–TCO reaction pair and azide–DBCO cycloaddition reaction pair have been used for multi-color labeling in live cells (Angew. Chem. Int. Ed. 2012, 51, 920–922; ACS Chem. Biol. 2021, 16, 2612–2622; Angew. Chem. Int. Ed. 2012, 51, 4166–4170; Bioconjugate Chem. 2020, 31, 5, 1370–1381) and multi-effector conjugation in antibody (Nat Commun. 2021, 12, 3528; Bioconjugate Chem. 2019, 30, 2452–2457) selectively and simultaneously.

In our own hand, as shown in Fig. 2c, one-pot labeling of Her2-scFv-pTAF with TCO-FITC and DBCO-CY5 resulted in only one product with a MS peak corresponding to the expected dual labelled product. Under our experimental conditions, we did not observe any peak representing dual FITC (Her2-scFv-K42-FITC-FITC: Expected Mass 29058.02 Da) or dual CY5 labeling (Her2-scFv-K42-CY5-CY5: Expected Mass 29390.76), suggesting that the one pot tetrazine-TCO IEDDA and azido-DBCO SPACC reactions are indeed mutually orthogonal for protein dual labeling containing pTAF. Overall, our results are consistent with these literature report. As you suggested, explanation about the orthogonality with proper references cited were added in the revised manuscript.

6. Authors chose A121 of J591 Fab for the incorporation of pTAF. The authors should explain why this residue was chosen.

Reply:

The site was chosen on the basis of its surface accessibility, high expression yield and high conjugation efficiency as demonstrated previously (Proc. Natl. Acad. Sci., 2012, 109, 16101-16106; Nat. Biotechnol. 2008, 26:925–932; Chem. Biol. 2011, 18:299–303). The yield of this mutant was similar to the yield of wild-type Fab under the same expression conditions.

7. The authors wrote that ^{64}Cu is labeled after the conjugate J591Fab-A121-pTAF with DBCO-CY5 and TCO-NOTA is formed, which does not match Fig. 4a showing that the TCO- ^{64}Cu is used in the first place instead of formed later. Also, the superscript of copper in Fig. 4a is in the wrong position.

Reply:

The Fig. 4a has now been corrected.

8. The molecular weight of the marker should be labeled in Figure 5b, at least for the ones have close MW to the protein band.

Reply:

The molecular weight of the marker has been labeled in Figure 5b.

9. Figure 2c & 6b the “scfv” should be changed to “scFv” to be consistent with the others in the paper

Reply:

All the “scfv” has been replaced with “scFv” in the revised manuscript.

10. In line 524, there an extra “)” should be removed.

Reply:

The extra “)” has been removed.

11. In line 668, a space between “CY5” and “(10 mM” is needed.

Reply:

The space has been added.

12. The figure quality is not good. High resolution figure is required.

Reply:

All high-resolution figures have been provided according to the requirements of Nature Communication.

13. “pTAF and mTAF” in the abstract should be spelled out.

Reply:

Thank you for the suggestion. 4-(6-(3-azidopropyl)-s-tetrazin-3-yl) phenylalanine (pTAF) and 3-(6-(3-azidopropyl)-s-tetrazin-3-yl) phenylalanine (mTAF) have been spelled out according to your suggestion.

Comments from Review #2:

The paper is devoted to the production and application of protein multiconjugates using a novel type of noncanonical amino acid (ncAA), which simultaneously contains in its chemical structure two bioorthogonal groups for site-specific labelling (azide and tetrazine). It has been shown that such ncAAs can be incorporated into the protein sequence using a genetic code expansion (GCE) technology and expressed proteins,

The authors present a new strategy for bioorthogonal protein labeling via an incorporation into a protein a newly synthesized ncAA, which is equipped with 2 functional groups taking part in bioorthogonal reactions: azide group – in strain-promoted alkyne-azide cycloaddition (SPAAC)

and tetrazine group – in inverse electron-demand Diels-Alder reaction (iEDDA). GCE technology was used to introduce such modified ncAA into the protein. Developed ncAA could provide an advantage in GCE efficiency, since it requires the incorporation of only one ncAA instead of two ncAAs, bearing each only one biorthogonal handle; and afterwards higher GCE efficiency leads to the increase in protein yield. Newly produced bi- and tri- conjugates were further labelled with dyes, radioisotopes and drugs in a “one-pot” manner and interesting applications were shown.

The introduction summarizes in detail and at length different strategies for making dual ncAA conjugates and highlights the disadvantages of dual encoding, which for GVVE typically requires to orthogonal RS pairs and suffers from lower expression yield. These are valid points that their method does not necessarily suffer from.

Reply:

We really appreciate your recognition of our work!

However, the introduction does not mention that dual functional group encoding ncAAs have been encoded more than once before. E.g. see *Chembiochem*. 2017 Jan 17;18(2):181-184 and or *Biochemistry*. 2018 Aug 7;57(31):4747-4752. Here, two other functional groups were encoded, one for labelling and for crosslinking, but from the type of journal published, it is also clear that not any combination of two known functionalities (of over 200 encoded ones) into one ncAA will depict a substantial advance.

Reply:

Thanks for your comments. Indeed, as you mentioned, all these bifunctional ncAAs reported have similar structural composition, which carries a click reaction handle and a photo-crosslinker (benzophenone or diazirine). They were exclusively used for proteomic purpose by photo-crosslinking and biotin attachment for pulling down protein complexes. On the contrary, encoding ncAA containing just one reactive group involved in inverse-demand Diels-Alder cycloaddition reactions were published in high profile journals (*J. Am. Chem. Soc.* 2020, 142, 7245–7249; *Angew. Chem. Int. Ed.* 2019, 58, 15876 – 15882; *J. Am. Chem. Soc.*, 2018, 140, 4860-4868.; *Angew Chem Int Ed Engl.* 2018, 5, 57(11):2831-2834; *J. Am. Chem. Soc.* 2015, 137, 10044–10047; *Nature chemistry*, 2014, 6(5): 393-403).

Protein therapeutics are gaining more and more attention in the field of translational medicine and site-specific protein conjugates have repeatedly demonstrated superior overall pharmacological profiles compared to their heterogeneous counterparts. The importance of site-specific protein therapeutics has been recently reinforced by a variety of examples in the development of site-specifically modified protein therapeutics (e.g. *Nat Biomed Eng* 2021,5, 1288–1305; *Nat Commun* 2021,12, 4785. Clinical Trials NCT04829604 and NCT04009681). However, as the need for multi-functionalization of protein increases, for example, combinations of drugs, fluorophores, radionuclides, PEG/materials, nucleic acids, immunotoxins, enzymes, et. al, there is an urgent need to develop methods allowing efficient, controllable, and scalable protein dual/multi-modifications. Although site-specific incorporation of two distinct ncAAs are possible for protein dual labeling, compared to one ncAA incorporation, the yields are too low for *in vivo* applications (**please find data and references in the following questions**). To this end, we have designed these dual-reactive amino acids (TAFs) containing mutually bio-orthogonal tetrazine and azido chemical groups into one side chain, and genetically encoded them into both *E. coli* and mammalian cells with high efficiency allowing preparation of recombinant proteins containing TAF site-specifically. Using antibody fragments as examples, we demonstrated that a combination of commercially available functional groups can be labeled simultaneously and site-specifically, including radionuclides (for early tumor diagnosis through PET imaging), fluorophore (for tumor surgical navigation through NIR imaging), PEG (for improved pharmacological properties) and drugs (for targeted cancer therapy). As we demonstrated, the yield of purified doubly conjugated proteins was high enough to afford *in vivo* experiments in several mouse models for above mentioned applications. Finally, we developed a system to simultaneously incorporate two ncAA, mTAF and another ncAA containing mutually exclusive bioorthogonal reactive group, such as ketone, into one protein using two stop codons, to afford a site-specific, triple-labeled protein in relatively good yield.

We believe that such ncAA containing mutually exclusive bioorthogonal reactive groups of azide and tetrazine will depict a substantial advance in the preparation of protein therapeutics and allowing efficient and high yield generation of many novel protein therapeutics, such as site-specific antibody drug conjugates containing two different payloads (toxins, sting agonists, TLR agonists, PROTACs, radionuclides, fluorophores, et. al.) using commercially available reactants. To differentiate our work with the previous bifunctional amino acids for proteomics purpose, we

have revised the title of work to “One-pot plug-and-play preparation of homogeneous protein multiconjugates by encoded doubly bio-orthogonal noncanonical amino acids”.

So overcoming the actual drawback of having two functional groups in close proximity (i.e. on one side chain) should be an important point to address for the readership of a broad interdisciplinary journal like Nature communication. The fact that no large linker can be encoded between two functional groups with current RS systems, and thus steric hindrance effects can become a concern, is a disadvantage that needs to be clearly shown to outcompete the disadvantages from dual ncAA incorporation with the discussed methods.

In that context, much more convincing needs to be done to make this relevant for the readership of Nature com in my eyes, such as- unfortunately, no side-by-side experiment has been demonstrated to compare how the presented approach outperforms a strategy with incorporation of 2 ncAAs. However it would be important to see such results, keeping in mind that new binding pockets for 2 synthetases (MjTyrRS and MbPylRS) was evolved for new ncAAs and efficiency of tRNA aminoacylation can be diminished.

Probable steric hindrance that might appear when bulky compounds are selected for protein modification. Steric hindrance theoretically can cause a decrease in a reaction rate or even absence of reaction when labelling group is shielded by a compound, which has been first covalently attached to a protein. Authors provide data in Fig. 2d, where they show that reactivity of tetrazine group is not compromised after reaction between DBCO-amine and azide group, however iEDDA reaction is several orders of magnitudes faster than SPAAC and tetrazine group will react first. Thus, it would be interesting to see the opposite experiment where tetrazine group has already reacted with a corresponding compound via iEDDA and then assess a reaction rate of SPAAC between azide group and complementary compound. If reaction rate of SPAAC is decreasing substantially, it might be a problem for conjugate production.

Reply:

Thank you for the comments. We have experimentally proved that “steric hindrance” is not a concern and genetic incorporating one ncAA confer substantial advantages in terms of protein yield and sites of selection over two different ncAAs for protein dual conjugates. Please find data in the following paragraphs. In fact, it would be very difficult to prepare protein dual conjugates

with sufficient quantity for applications in animals by simultaneous incorporating two different ncAAs into one protein. We are not saying that it's not possible, but it certainly requires large trials and errors to identify two mutating positions that both give high expression yield and do not interfere with protein folding and functions upon conjugation. We think competing with release factors twice for two different stop codons really have a substantial impact on the protein yield regardless of synthetase efficiency. This was confirmed by the fact that no literature has ever reported animal studies using one recombinant protein containing two different ncAAs. Experimental data to address your concern has been provided as follow.

“steric hindrance effect”

First, regarding your concern about “steric hindrance effect”, we have determined the reaction rates of azide in the presence of unreacted tetrazine or tetrazine occupied after reacting with TCO-small molecule or TCO-PEG20K at both the free amino acid level and proteins containing pTAF residue, respectively. The results verified that no obvious steric hindrance effect on azide reactivity was observed on pTAF at both amino acid and residue levels, indicating that the propyl linker between azide and tetrazine is long enough to afford one-pot dual reactions.

1: Free amino acid

To begin with, compounds pTAF-Biotin (**2**, Figure R4) and pTAF-PEG20K (**3**, Figure R4) were prepared by mixing pTAF amino acid (**1**, Figure R4) with TCO-PEG4-Biotin or bulky TCO-PEG20K with 1:1 stoichiometry, respectively, followed by purification using semi-preparative HPLC and lyophilization.

The reaction rate constants of free pTAF, pTAF-Biotin and pTAF-PEG20K with DBCO-COOH, were determined using HPLC analysis at 25 °C. Product peaks were detected by 2998 photodiode array detector at 254 nm (Waters), the peak area of reactants and products were taken at different time points over a period of 12-24 h using a 10-50% B gradient 10 min and 50-10% B gradient next 10 min. Peak areas were integrated for relative quantification using the Empower software (Waters). The second order rate constants of free pTAF, pTAF-Biotin, and pTAF-PEG20K with DBCO-COOH were then determined to be **$0.70 \pm 0.4 \text{ M}^{-1}\text{s}^{-1}$** , **$0.66 \pm 0.03 \text{ M}^{-1}\text{s}^{-1}$** and **$0.30 \pm 0.02 \text{ M}^{-1}\text{s}^{-1}$** , respectively (Figure R4). The results indicate that smaller modification group TCO-PEG4-Biotin has no steric effect. Although the bulky modification PEG20K reduced the reaction rate by

two-fold, the rate is still within the same order of magnitude, which is sufficient for protein dual labeling. These data have been included in the revised manuscript and Figure S10.

Figure R4. Measured second-order rate constants of free pTAF, pTAF-Biotin, and pTAF-PEG20K with DBCO-COOH respectively in MeOH at 25 °C. Note that the different ordinates were chosen for integrating based on the different UV absorption strengths of the reactants or products. The total peak area of reactants or products were added and plotted against time and calibration curves were constructed to convert peak area at each spot into $1/c - 1/c_0 = kt$ (c : concentration of reactants at a certain time; c_0 : the initial concentration of reactants), $1/(c_0 - c_p) - 1/c_0 = kt$ (c_0 : the initial concentration of reactants; c_p : concentration of product at a certain time), OR fit to one phase exponential decay using the Prism software package. Three independent trials were averaged by using the standard deviation as the error. Note: No error bars appear for some points because they are shorter than the size of the symbol.

2: TAF residue on protein

IFN- α 2b containing pTAF at the site of E42 was chosen as a model protein. To occupy the tetrazine position, TCO-PEG3-FITC and TCO-PEG20K were reacted with the tetrazine group of IFN- α 2b-E42pTAF, respectively and the resulting protein conjugates were purified to obtain IFN- α 2b-PEG3-FITC and IFN- α 2b-PEG20K. **The rate constant of azide on IFN- α 2b-pTAF, IFN- α 2b-PEG3-FITC and IFN- α 2b-PEG20K with DBCO-CY5 were determined to be $181.66 \pm 31.32 \text{ M}^{-1}\text{s}^{-1}$ (Figure R5a) $746.27 \pm 125.78 \text{ M}^{-1}\text{s}^{-1}$ (Figure R5b) and $177.60 \pm 11.18 \text{ M}^{-1}\text{s}^{-1}$ (Figure R5c), respectively.** Interestingly, the reaction rate of azide on IFN- α 2b-PEG3-FITC, where the tetrazine was occupied by TCO-PEG3-FITC, was four-fold higher than the one on IFN- α 2b-pTAF, where the tetrazine was unoccupied. It is possible that the π - π interaction or hydrophobic interaction between fluorescein and CY5 could accelerate the reaction. **Most importantly, the reaction rate of azide on IFN- α 2b-PEG20K ($177.60 \pm 11.18 \text{ M}^{-1}\text{s}^{-1}$) was identical to the reaction rate of azide on IFN- α 2b-pTAF ($181.66 \pm 31.32 \text{ M}^{-1}\text{s}^{-1}$), indicating that the bulky PEG20K attaching the tetrazine group does not have a steric hindrance effect on the following azide-DBCO reaction, which I hope can address your concern.** It is worth mentioning that the rate constants measured on proteins are significantly faster than those on the free amino acid. This is due to the relative polarity between the protein surface and buffer, as reported previously (J. Am. Chem. Soc. 2020, 142, 7245–7249). These data have been included in the revised manuscript and Figure S11.

Figure R5. Second-order rate constants determination between azide and DBCO-CY5 on IFN-α2b-pTAF (a), IFN-α2b-PEG3-FITC (b) and IFN-α2b-PEG20K (c). The reaction rates were determined by time course fluorescent gel imaging using Typhoon. All measurements were repeated three times and the error bars represent the standard deviation. Note: No error bars appear for some points because of they are shorter than the size of the symbol.

“Protein yields”

Second. the yield of recombinant protein mutant containing ncAA has always been a concern for large scale protein production and industrial applications. Now, as you mentioned, after years of optimization, it is now possible to obtain a mutation site with a ncAA containing mutant protein yield comparable to the wildtype protein through mutation position screening using optimized synthetase and expression system. **However, the efficiency for dual ncAAs incorporation is still way too low and becoming the biggest shortcoming if sufficient amount of protein is required**

to perform animal studies. This was supported by the fact that no dual ncAAs incorporation has been reported for animal studies. At least one reason is that for each ncAA incorporation, there is a competition between stop codon suppression and ribosome release. The efficiency decreases dramatically after two competitions for two different stop codons. Although the recent development of RF1-knockout strains may overcome the TAG codon competition with release factor 1, the strategy does not apply for two different ncAA incorporation using two different stop codons. Besides, these RF1-knockout strains grow much slower than the commonly used BL21 strain and resulted in a much less recombinant protein production yield. For example, Ryan A. Mehl lab encoded ncAAs pAzF and Tet 3.0 containing aromatic azide group and tetrazine group, respectively, into one protein (SUMO-sfGFP) simultaneously and achieved dual modification of proteins (ACS Chem. Biol. 2021, 16, 2612–2622). However, the yield of expression is only about 10% of wt SUMO-sfGFP. It worth to mention that the SUMO-sfGFP is a highly expressed model protein, and based on our own experience, the yield decreases substantially when purifying low expressing proteins, such as antibody fragment or cytokines. This is because for proteins with low expression yield, the binding to the affinity resin decreases and non-specific binding increases considerably, which often require a second purification step to obtain relatively pure protein and further decreased the final protein yield.

To answer your question, we have performed our own side-by-side experiment using these two reported ncAAs (pAzF and Tet 3.0, Figure R6a) according to Dr. Mehl's work. In their work, the SUMO-sfGFP fusion is an artificial two domain protein and they incorporate the two ncAAs on two different domains. As each domain is individually folded, we think it's better to incorporate both ncAA onto a single domain protein to reflect our needs to modify therapeutic proteins. To evaluate the expression, sfGFP possessing pAzF at a most selected site of 151 and Tet3.0 at a few solvent accessible sites was expressed and the efficiency for incorporation was evaluated by the normalized fluorescence intensity. As shown in Figure R6b, the yield of protein mutants containing two ncAAs ranging from 2% - 10% as compared to protein mutant containing only one ncAA. While, putting the release factor competition aside, we think it is certainly possible to further increase the yield of double incorporation by screening more mutation site combinations and optimizing the synthetases, cell strains, and expression cassettes. However, there is no doubt that it would significantly increase the amount of workload and making the process very inconvenient.

Figure R6. **Quantification of sfGFP mutant expression using sfGFP stop codon suppression assay.** (a) Structures of ncAAs used in this study. (b) normalized fluorescence intensity of sfGFP. All measurements were repeated three times and the error bars represent the standard deviation.

Based on the data of screening mutation sites for dual incorporation, we chose two pairs of sites at 89TAA/151TAG and 117TAA/151TAG to express the corresponding double mutant proteins in *E. coli*. Subsequently, dual labeling experiment was performed on sfGFP-151pTAF, sfGFP-151pAzF-89Tet3.0 and sfGFP-151pAzF-117Tet3.0 with DBCO-ssDNA and TCO-PEG5K, respectively. Interestingly, the conjugation efficiency for both sfGFP mutants containing two ncAAs was much lower than sfGFP mutant containing our single ncAA pTAF for dual conjugation (Figure R7). It is likely due to the arylazide reduction and the slow reaction rate of pAzF (ACS Chem. Biol. 2021, 16, 2612–2622; Org. Lett. 15, 4442–4445; Cell Chem. Biol. 2016, 23, 805–815), which further compromise the labeling reactions. The problem could potentially be avoided by using alkyl azide amino acid, however, the development of a highly efficient and mutually orthogonal dual ncAA incorporation system for alkyl azide and tetrazine ncAAs is beyond the scope of our work.

Overall, we believe that the data is very convincing that our TAF amino acids do not have the steric hindrance problem as you concerned and confer substantial advantages in terms of protein expression yield, site of selections and dual labeling efficiency, which makes our methods the

state-of-the-art choice for site-specific protein dual labeling for both convenience and high yields. We will upload our plasmids onto the addgene website, where we already have a group page, and make the method largely accessible to the scientific community. We wish that more combinations of different chemicals onto antibodies and therapeutic proteins can be explored on animal models in the future by using our approach.

Figure R7. Labeling efficiency of sfGFP-151pTAT containing azide and tetrazine on a single ncAA, and sfGFP-151pAzF-89Tet3.0 and sfGFP-151pAzF-117Tet3.0 containing azide and tetrazine on two separate ncAAs. Protein mutants were reacted with DBCO-ssDNA and TCO-PEG5K, respectively.

I have two more major concerns, that are easier addressed but currently lower my enthusiasm for the paper.

- Cy5 is a red fluorescent dye, and not an IR dye. I checked some marketing flyers from some producing companies, and even those do not claim Cy5 to be near IR. Cy5.5 is the first dye that is considered by those companies with self-interest as near IR, and I would argue that from a visible spectrum point of view, the first proper IR dye from the cyanine derivatives is Cy7.

Reply:

This is a good question. The fact that some articles reported the emission wavelength for CY5 as 670 nm whereas others reported as 700 nm. Therefore, some articles did not consider CY5 as a near-infrared dye because they defined 700–900 nm wavelength range as near-infrared (Nature communications, 2020, 11(1): 1-11), but there are still some published works consider CY5 a near-infrared dye because they defined the wavelength of near-infrared as 650–950 nm (J. Am. Chem.

Soc. 2012, 134, 13730–13737; Angew. Chemie. Int. Ed. 2021, 133, 17026–17030. Front. Bioeng. Biotechnol. 2020, 7, 487). We would argue that the definition of CY5 as a near IR dye is ambiguous. Nevertheless, we agree with you that the tumor imaging study should use CY7 as a proper IR dye. In fact, that’s exactly what we were doing during the submission of this work. The reason we used CY5 dye at first was due to the convenience of fluorescence imaging to characterize the binding of dye labeled antibody fragment to the PCa cell lines, as CY7 is beyond the detection limit of these instruments. We included CY5 animal data in the manuscript for the purpose of quick submission, while we were waiting for the CY7 data. Since now we have obtained the new tumor imaging data using CY7 labeled antibody fragments, we have replaced the CY5 data in the revised manuscript and Figures 3, S16 and S17 to avoid any concerns.

Figure R8. Preparation of anti-PMSA antibody fragment bearing site-specifically labelled near-IR dye-CY7 and PEG dual conjugate for imaging of PMSA positive tumor with optimal time window in mice. (a) In vivo whole-body NIRF imaging of PCa-bearing mice after intravenous injection of J591Fab-CY7 or J591Fab-CY7-PEG20K at various time points. The dotted circles indicate tumor locations. (b) Quantification of fluorescence intensity of 22Rv1 tumors in panel a; n = 4 mice for each group. (c) Ex vivo evaluation of dissected organs at 72 h after injection of J591Fab-CY7 and J591Fab-CY7-PEG20K. Organ labels: 1, heart; 2, liver; 3,

spleen; 4, lung; 5, kidney; 6, stomach; 7, muscle; 8, small intestine; 9, large intestine; 10, tumor. (d) Tumor-to-background ratios for 22Rv1-tumor-bearing mouse in panel c. Muscle was used as the background tissue; n = 4 mice for each group. Mann-Whitney U test ($p = 0.029$). Data in b and d are means \pm SDs.

- I appreciate the breath of experiments done for important applications of multi conjugates for diagnostic and therapy and thus make the paper very interesting to read indeed, However, the statistics in many and in particular the mice experiments are low and I understand that one needs to be considerate with number of mice sacrificed and I do not suggest to do more of those in the interest of living mice. However, at the same time, how this is sold needs to be also careful. The students t-test is not even defined for a sample size of 3 or 4, and the use of proper statistical tools or more careful wording is certainly needed when advertising the work.

Reply:

We really appreciate your recognition of our work and your admirable humane criteria for animal. We have changed the statistics used in this study to nonparametric tests, which were more suitable for small sample size statistic than t test (Siegel, S. Nonparametric statistics. The American Statistician, 1957, 11, 13-19.; Elliott, A. C., & Woodward, W. A. (2007). Comparing one or two means using the t-test. Statistical Analysis Quick Reference Guidebook. SAGE.). Mann-Whitney U test was used to compare the differences between two independent samples. Kruskal-Wallis test and Tukey HSD test were used to compare the differences between three or more independent samples.

1. The statement in line 447: “Taken together, our results show that antibody fragments site-specifically modified by dual-reactive pTAF could be used to prepare homogeneous fragment-PEG-drug dual conjugates” need its own set of analytics to validate the homogenous claim.

Reply:

Thank you for the careful review of our paper. For a side-by-side analysis, we labeled scFv-pTAF with TCO-PEG20K to generate scFv-PEG20K, and simultaneously labeled scFv-pTAF with DBCO-MMAE and TCO-PEG20K to generate scFv-MMAE-PEG20K in one pot. The reaction was near complete by SDS-PAGE analysis (Figure R9a). The two conjugates were further analyzed by the MALDI-TOF MS, and the signals of scFv-PEG20K (Green shaded area, Figure

R9b) and scFv-MMAE-PEG20K (Blue shaded area, Figure R9b) were completely separated in MALDI-TOF MS spectra, which demonstrates that our dual conjugates contain not only PEG20K but also MMAE coupled. However, since the PEG20K molecule itself is not a homogenous species, to avoid any confusion, we have revised the claim to “...used to prepare site-specifically PEGylated homogeneous antibody fragment-drug conjugates.”

Figure R9. SDS-PAGE and MALDI-TOF MS analysis of scFv-PEG20K and scFv-MMAE-PEG20K.

2. Statement line: 268 filtration (~60 kD)⁴⁹. It was small enough for enhanced tumor penetration and it had a faster serum clearance rate than full-length antibodies” needs reference or experiment.

Reply:

IgG (~150 kDa) has a slow clearance rate (weeks/days) due to salvage recycling by the neonatal Fc receptor (FcRn) pathway and exclusion from renal filtration (cut-off ~60–70 kDa) (Antibodies 2018, 7, 16). Furthermore, the binding site barrier also limits its penetration in solid tumors (Cancer Res, 1989, 49, 20:5656–63; Cancer Res, 2021, 81, 15: 4145–4154). Pegylated Fab-20kDa would

have the potential to overcome the shortcomings of full-length antibodies for enhanced tumor penetration and a faster serum clearance of Fab, and this is an ongoing project of our anti-tumor therapy development. We have revised the claim to “It would have the potential for enhanced tumor penetration and better in vivo pharmacokinetics”.

3. Page 5. “anhydrous hydrazine, which is ... and not commercially available”. I would recommend to omit this detail here.

Reply:

Thank you for the suggestion. This description has been deleted.

4. Page 6. “218-peptide linker (GSTSGSGKPGSGEGS)”. Please cite a paper, from which sequence of linker was taken. At least in Whitlow M. et al, Protein Eng., 1993 the sequence of 218-peptide linker is GSTSGSGKPGSGEGSTKG.

Reply:

Thank you for the careful review of our paper. Indeed, both sequences were used as 218 linker in the reference. “GSTSGSGKPGSGEGS” was used a few references, for examples Nat. Commun, 2011, 2: 406, Molecular therapy, 2020, 28, 2, 536-547, and US 2019/0092818 A1. The difference between the two types of 218 linker is that the second one contains additional “TKG” residues at the end of the “GSTSGSGKPGSGEGS” sequence, which does not seem to have an impact on antibody fragment expression and function.

5. Figure S5. In the caption it is mentioned that protein migrates as a single band at about 27 kDa, but 2 bands are clearly visible on the gel. Please use accurate description and do you have any assumptions why 2 bands appeared?

Reply:

The protein was not pure enough. We re-expressed and purified the two proteins. The Figure S5 has been replaced with the new SDS-PAGE gel.

Figure R10. SDS-PAGE analysis of WT Anti-HER2-scFv and Anti-HER2-scFv-K42-pTAF mutant. Both proteins were purified from *E. coli* periplasmic space and migrated as a single band around 27 kDa.

6. Fig. 1c and S5 uses different style: “Kd” and “kDa”. Please unify the style of labeling.

Reply:

Thank you for the careful review of our paper. We have changed the “Kd” and “kDa” into “kDa” for uniformity thorough the manuscript.

7. Fig. S8 – please insert a title of the caption.

Reply:

Thank you for your reminder. We have inserted the title of the caption in Fig. S8 as “Plotting kinetics curve of $1/c-1/c_0$ with time for sfGFP-pTAF- N_3 (N_3 unreacted) and sfGFP-pTAF- N_3 -DBCO-amine (N_3 occupied)”.

8. Fig. S11. Why was FRET channel omitted in this figure? For other figures demonstrating results for in-gel fluorescence this channel was demonstrated.

Reply:

FRET fluorescence has now been included in the revised Figure S13.

Figure R11. **Fluorescence SDS-PAGE images of conjugate J591 Fab-pTAF upon treatment with DBCO-CY5 and/or TCO-PEG3-FITC.** Denatured and reduced 12% SDS-PAGE gel. Coomassie blue protein stain. Fluorescence conjugations were visualized by fluorescence imaging using a Typhoon Imager.

9. In Fig. 3d another bright focus of dye accumulation rather than the tumor is visible. Does it belong to accumulation in kidneys or stomach? Interestingly, the brightest organ after dissection seems to be a stomach according to Fig. 3f. Fig. S15 shows that side dye accumulation happens preferentially in kidney and stomach, while text says: “Ex vivo evaluation of dissected organs at 72 h p.i. revealed that fluorescence appeared mainly in the liver and kidneys”. Please fix the text to avoid misunderstanding.

Reply:

Thank you for pointing it out. Considering that the kidney showed the highest uptake in the nuclear imaging and biodistribution result in Figure 4c and 4f, we believe that the bright focus in Fig. 3d should belong to the accumulation in the kidney. We identified that the high fluorescence signal in the stomach come from the food source, which showed bright fluorescence in both the CY5 and CY7 channels. To reduce this food fluorescence background, we removed food debris from mouse

stomach prior to fluorescence imaging of dissected organs and the update the figures (Figure 3f) in the revised manuscript.

Figure R12. **The fluorescence imaging of food and dissected organs in the CY5 and CY7 channel.** Dissected organs of a PBS-treated mouse were scanned with the IVIS imaging system. Organ labels: 1, heart; 2, liver; 3, spleen; 4, lung; 5, kidney; 6, stomach; 7, small intestine; 8, large intestine; 9, muscle.

10. “J591 Fab containing a pTAF mutation at A121.” That is not clear what “pTAF mutation” is. Please rephrase this sentence.

Reply:

We have rephrased the sentence to “J591 Fab containing a A121 pTAF mutation, where the alanine at 121 position was mutated to ncAA pTAF by site-directed mutagenesis of alanine codon to an amber codon....”

11. “The apparent binding affinities (EC₅₀) of scFv-CY5, scFv-CY5-PEG20K, and scFv-CY5-PEG40K were determined to be 1.163, 7.316, and 23.93 μ M, respectively (Fig. S19). These results indicate that the conjugates had high binding affinity for the HER2 receptor, although the PEGylation did have influence the binding in a size-dependent manner.” Measured binding affinities do not seem to be high, taking also into account that Her2-scFv with higher nanomolar K_d have been published. Did you try another method to check and calculate binding affinity of your wild-type Her2-scFv?

Reply:

Thank you for the suggestion. Indeed, the apparent binding affinity between HER2-scFv and its receptor on HCC1954 breast cancer cells was estimated by flow cytometry to have an estimation about the effect of PEGylation. That's why we used the term "apparent binding affinities (EC50)" instead of KD value. As you suggested, we determined the true binding affinity between HER2/ERBB2 protein and scFv-WT (Figure R13, a), scFv-pTAF-CY5 (Figure R13, b), scFv-pTAF-PEG20K-CY5 (Figure R13, c), and scFv-pTAF-PEG40K-CY5 (Figure R13, d) by means of biolayer interferometry, and the KD values were determined to be 17.0 ± 1.6 pM, 72.1 ± 1.3 pM, 0.22 ± 0.005 nM, and 0.45 ± 0.007 nM for each protein conjugate, respectively. This result confirmed your comments that the binding was indeed very strong (Table R1). The data has been included in the Figure S21 and Table S1 in the revised manuscript.

Figure R13. Biolayer interferometry analysis of HER2/ERBB2 protein with scFv-WT (a), scFv-pTAF-CY5 (b), scFv-pTAF-PEG20K-CY5 (c), and scFv-pTAF-PEG40K-CY5 (d). The experimental detail can be found in the method section. BLI experiments were performed on an Octet RED96e instrument (ForteBio Inc., Sartorius, Germany) using amine-reactive AR2G

biosensors. Sensor tips were hydrated for 10 min prior to use. The sensors were then activated with a freshly prepared mixture of 90 mM EDC and 25 mM sulfo-NHS for 5 min, loaded with 20 $\mu\text{g/mL}$ HER2/ERBB2 Protein in 10 mM sodium acetate (pH 4.5) for 5 min, and then excess reactive esters were quenched with 1 M ethanolamine, pH 8.5 for 5 min. Amine-coupled HER2/ERBB2 Protein was then used to capture a panel of scFvs (scFv-WT, scFv-CY5, scFv-PEG20K-CY5 and scFv-PEG40K-CY5) in kinetics buffer (PBS with 0.05% Tween20). Association was monitored for 5 min and dissociation was monitored for 20 min. Each kinetics experiment included the analyte scFv in kinetics buffer at several different concentrations, as well as a reference biosensor, loaded with HER2/ERBB2 Protein, which was kept in kinetics buffer without antibody as a zero concentration. Octet BLI Analysis 12.2 software was used for the analysis of the binding curves.

Table R1: The affinity (KD , $k_{\text{off}}/k_{\text{on}}$), the association-rate (k_{on}) and dissociation-rate (k_{off}) of scFv-WT, scFv-CY5, scFv-PEG20K-CY5 and scFv-PEG40K-CY5

	KD (M)	k_{on} ($\text{M}^{-1} \text{s}^{-1}$)	k_{off} (s^{-1})
scFv-WT	$(1.70 \pm 0.16) \times 10^{-11}$	$(2.03 \pm 0.003) \times 10^5$	$(3.45 \pm 0.31) \times 10^{-6}$
scFv-CY5	$(7.21 \pm 0.13) \times 10^{-11}$	$(1.89 \pm 0.003) \times 10^5$	$(1.36 \pm 0.02) \times 10^{-5}$
scFv-PEG20K-CY5	$(2.20 \pm 0.05) \times 10^{-10}$	$(6.09 \pm 0.01) \times 10^4$	$(1.34 \pm 0.03) \times 10^{-5}$
scFv-PEG40k-CY5	$(4.52 \pm 0.07) \times 10^{-10}$	$(5.45 \pm 0.01) \times 10^4$	$(2.46 \pm 0.04) \times 10^{-5}$

12. Fig. 5C. It seems not to be any colocalization between lysotracker and scFv-CY5-PEG40K, while text says: “These results demonstrated that the scFv-CY5-PEG conjugates could enter the lysosome degradation pathway after internalization”. Please fix the test or describe more in details.

Reply:

Thanks for the comments. We have provided new data in the revised manuscript. Please refer to Reviewer 1 question 1 for details.

13. “All the conjugates reduced HCC1954 cell viability, and the EC50 values were calculated to be 9.7, 380.6, and 360.3 nM for scFv-MMAE, scFv-MMAE-PEG20K, and scFv-MMAE-PEG40K, respectively”. Probably authors meant here IC₅₀, but not EC₅₀. Please fix.

Reply:

The term “IC₅₀” has been used in the revised manuscript.

14. “The maximum intensity of tumors in scFv-CY5-PEG40K group ($\sim 2.10 \times 10^9$ p/sec/cm²/sr at 72h) was 5.13 folds higher than that in scFv-CY5 group ($\sim 4.09 \times 10^8$ p/sec/cm²/sr at 2h).” It would be better to use the same extent for values to make more clear 5.13-fold difference.

Reply:

The same extent for values had been applied in the revised manuscript.

15. Fig. S21 – ALT and AST abbreviations should be explained, they might not be obvious for all readers.

Reply:

Thanks. The definition, “ALT, Alanine aminotransferase; AST, Aspartate aminotransferase” have been explained and added in the revised manuscript and in Figure S21. (Current Figure S24 in the revised Supplementary information).

16. Fig. 7b. Gel shows a dim, but clear band for “-ncAA”. Do you suppose to have leakage in orthogonality there? Did new synthetase start recognizing endogenous AA? Did you do any negative selection to prevent recognition of endogenous ncAA?

Reply:

Thank you for the careful review of our paper. Yes, in the absence of ncAA, there is a slight leakage for natural amino acid incorporation at the non-sense codon position in rich medium, such as 2YT. However, no background incorporation can be observed for protein expressed in minimal medium, such as M9, a condition used for negative selection (Figure R14a). Based on our experience, as well as literature reports (Biochemistry 2013, 52, 1828–1837), there is a balance between orthogonality and protein yield. In fact, for many synthetases or expression systems to afford high protein yield, a leaky expression can often be observed in the absence of ncAA, especially using rich medium. The background incorporation is usually large hydrophobic amino

acids, such as Phe and Trp. To answer your question, we analyzed the “dim, but clear band” (Figure R14, line 2) of protein mutant expressed in 2YT and in the absence of mTAF by high-resolution electrospray-ionization mass spectrum. The result revealed that phenylalanine and tryptophan were inserted into the amber codon site of 151-TAG (Figure R14, b).

However, such leaky expression is not a concern at all. When the corresponding ncAA is provided, the background incorporation can be fully suppressed, and the fidelity of incorporation of the ncAAs into the target protein is high (Biochemistry 2013, 52, 1828–1837). To prove this, sfGFP containing a 151-TAG was expressed in the presence of mTAF in both 2YT medium and M9 medium. High-resolution electrospray-ionization mass spectrum on purified protein confirms that only a single peak corresponding the correct incorporation of mTAF can be observed (Figure R14, c, d). No peak corresponding to the incorporation of natural amino acids can be observed suggesting that the leakage expression can be fully suppressed in the presence of ncAA, which is consistent with the previous report (Biochemistry 2013, 52, 1828–1837). For therapeutic protein, the high yield expression is one of the most critical concerns. Even though the background leaky expression in the absence of ncAA is often observed in such high expression systems, based on our MS results and literature report, it should not be a concern when the mutant protein is expressed in the presence of ncAA. The resulting protein is nearly 100% pure with ncAA mutation.

Figure R14. SDS page and high-resolution electrospray-ionization mass spectrum analysis of sfGFP mutant expressed in the absence or presence of mTAF in 2YT or M9 medium.

17. Fig. 7d. Sequence in blue tRNA should be UUA as mentioned in a text. Please fix a mistake.

Reply:

The mistake has been corrected in the revised manuscript in Figure 7d.

18. Fig. S26. Quite bright band can be observed for condition “- pAcF + mTAF”, so it seems that MjTyrRS starts recognizing mTAF for tRNA aminoacylation. At the same time there are also dim bands already in conditions “- pAcF - mTAF” and “+ pAcF - mTAF”. This evidence of leakage in orthogonality should be mentioned somewhere in a text or in the caption.

Reply:

This question is related to question 16. MjTyr-derived tRNA/aaRS pairs exhibit higher background suppression levels than their pyrrolysine counterparts (Biochemistry 2013, 52, 1828–1837). This

is consistent with the observation in SDS-PAGE analysis and High-resolution electrospray-ionization mass spectrum that in the absence of pAcF, *Mj*Tyr-derived tRNA/aaRS pairs exhibit higher background (Phe or Trp) suppression levels (Figure R15a, line 3, 7, Figure R15b, and Figure R15c) than their pyrrolysine background (Gly and two pAcFs) in the absence of mTAF (Figure R15a, line 2, 6, Figure R15d, and Figure R15e) no matter whether in 2YT or M9 medium. The leakage would be significantly reduced when *E.coli* was cultured in M9 medium. However, robust production and purity of GFP was observed only in the presence of both pAcF and mTAF (Figure R15a, line 1 and 5 and Figure R15f), with little protein expressed and obtained to purity in the absence of both pAcF and mTAF (Figure R15a, line 4 and 8).

Figure R15. Comparison and analysis of suppression efficiencies of mediums type and the type of ncAAs.

19. Fig. S27. Please precise what you mean with “-ncAA”: absence of 1 or 2 ncAAs.

Reply:

Yes, it means in the absence of ncAA. We have made the corresponding corrections in Figure S27 (Current Figure S30 in the revised Supplementary information).

REVIEWERS' COMMENTS

Reviewer #1 (Remarks to the Author):

The authors offered a detailed response to reviewers' comments and significantly improved the manuscript. The manuscript is worthy of being published in Nature Communications.

Reviewer #2 (Remarks to the Author):

This is a fine revision, the paper is now ready for publication.

The authors would like to thank the reviewers and editor for supportive comments with our manuscript. Herein, we address the comments of **NCOMMS-22-32799A** as follows.

Point-by-point response for **Manuscript NCOMMS-22-32799A**

Comments from Review #1:

The authors offered a detailed response to reviewers' comments and significantly improved the manuscript. The manuscript is worthy of being published in Nature Communications.

Reply:

We thank the reviewer for the very supportive comments.

Comments from Review #2:

This is a fine revision, the paper is now ready for publication.

Reply:

We really appreciate your recognition of our work.